# Federated analysis of autosomal recessive coding variants in 29,745 developmental disorder patients from diverse populations

V. Kartik Chundru [1,2], Zhancheng Zhang [3,4], Klaudia Walter [1],
Sarah J. Lindsay [1], Petr Danecek[1], Ruth Y. Eberhardt [1], Eugene J. Gardner [1,5],
Daniel S. Malawsky[1], Emilie M. Wigdor[1,6], Rebecca Torene [3,7], Kyle Retterer[3,7],
Caroline F. Wright [2], Hildur Ólafsdóttir[8], Maria J. Guillen Sacoto[3], Akif Ayaz[9],
Ismail Hakki Akbeyaz[10], Dilşad Türkdoğan[10], Aaisha Ibrahim Al Balushi[11],
Aida Bertoli-Avella [12], Peter Bauer[12,13], Emmanuelle Szenker-Ravi [14],
Bruno Reversade [14], Kirsty McWalter [3], Eamonn Sheridan[1,15,16],
Helen V. Firth[1,17], Matthew E. Hurles [1], Kaitlin E. Samocha [1,18,19,20],
Vincent D. Ustach [3,21] & Hilary C. Martin [1,21] ✉

Autosomal recessive coding variants are well-known causes of rare disorders. We quantified the contribution of these variants to developmental disorders in a large, ancestrally diverse cohort comprising 29,745 trios, of whom 20.4% had genetically inferred non-European ancestries. The estimated fraction of patients attributable to exome-wide autosomal recessive coding variants ranged from ~2–19% across genetically inferred ancestry groups and was significantly correlated with average autozygosity. Established autosomal recessive developmental disorder-associated (ARDD) genes explained 84.0% of the total autosomal recessive coding burden, and 34.4% of the burden in these established genes was explained by variants not already reported as pathogenic in ClinVar. Statistical analyses identified two novel ARDD genes: *KBTBD2* and *ZDHHC16*. This study expands our understanding of the genetic architecture of developmental disorders across diverse genetically inferred ancestry groups and suggests that improving strategies for interpreting missense variants in known ARDD genes may help diagnose more patients than discovering the remaining genes.

High-throughput exome and genome sequencing[1] have revolutionized the diagnosis of developmental disorders[2], typically allowing 30–40% of patients to obtain a genetic diagnosis[3–5]. Multiple new developmental disorder-associated genes have been discovered by statistical analysis of sequence data from large, phenotypically heterogeneous cohorts[6–11]. For example, a recent study brought together >30,000 trios, primarily from the Deciphering Developmental Disorders (DDD) study and the USA-based diagnostic testing company GeneDx, and identified 28 novel genes in which de novo mutations are likely to cause developmental disorders[6].

This contrasts with the more traditional approach of phenotype-driven gene discovery based on small numbers of patients or families who appear to have the same rare, clinically recognizable disorder (for example, Miller syndrome[12], Wiedemann–Steiner syndrome[13]) or the 'matchmaker exchange' approach in which researchers identify additional patients from other cohorts who have potentially damaging variants in the same candidate disease gene as an index patient[14].

The genetic architecture of developmental disorders has been shown to vary between genetically defined ancestry groups

as a result of varying levels of consanguinity[7,15]. In a study of 6,040 exome-sequenced patients from the DDD study, we previously estimated, through exome-wide burden analysis, that ~4% of probands with European ancestries and ~31% of those with Pakistani ancestries could be explained by autosomal recessive coding variants versus ~50% and ~30%, respectively explained by de novo coding mutations[7]. Only 48% of the exome-wide burden of recessive causes was in known ARDD genes, indicating that larger sample sizes and/or a different study design would be required to find the additional genes. Here, we combine a larger set of DDD trios with data from GeneDx to study the recessive contribution to developmental disorders in a set of 29,745 trio probands across 22 genetically inferred ancestry (GIA) groups, of whom 20.4% have majority non-European ancestries. We first quantify the recessive contribution to developmental disorders across GIA groups and the extent to which this is explained by known genes and known pathogenic variants. We next explore the contribution of multi-gene diagnoses using similar burden analyses. Finally, we conduct gene-based burden testing to identify new recessive genes underlying developmental disorders.

## Results

We analyzed deidentified exome-sequence data from the DDD study (*n* = 13,450 probands) and from GeneDx (*n* = 36,057). Given that the vast majority of DDD patients have at least one Human Phenotype Ontology (HPO) term under 'abnormality of the nervous system', we selected GeneDx patients with at least one such term for inclusion in this analysis. There were differences in the reported phenotype distributions between the cohorts (Supplementary Figs. 1 and 2 and Supplementary Table 1); however, these are likely to be largely attributable to the way HPO terms were recorded: DDD clinicians recorded HPO terms that they thought were particularly distinctive and likely to be relevant to a monogenic disorder, whereas in GeneDx, the HPO terms were abstracted from each patient's medical history (see Methods). Consequently, many of the terms that differed in prevalence between the cohorts were nonspecific and/or indicated common conditions (for example, 'failure to thrive', 'asthma') (Supplementary Note). Both cohorts have considerable heterogeneity of phenotypic presentations and genetic etiologies and have a similar burden of de novo mutations[6]. Given all this information, we decided that the two cohorts were sufficiently similar to combine them for the work in this paper.

We began by classifying individuals into GIA groups. The rationale for this classification was twofold. Firstly, we were interested in exploring differences in genetic architecture between these groups; secondly, the analysis below relies on accurate estimates of allele frequencies that differ between groups. We recognize that these GIA groups do not capture the full genetic diversity of human populations. We determined the GIA groups for each cohort in a federated manner, that is, by analyzing summary statistics that were produced without physically combining the individual-level genetic data. The classifications were based on genetic similarity to individuals in the 1000 Genomes and Human Genome Diversity Panel reference datasets, inferred from principal component analysis (Extended Data Figs. 1 and 2). Using statistical clustering of individuals based on their genotypes, we defined six continental-level GIA groups (AFR, African; AMR, Latin American; EAS, East Asian; EUR, European; MDE, Middle Eastern; and SAS, South Asian) and, within these, 47 fine-scale GIA sub-groups (Supplementary Table 2).

We carried out federated quality control on the exome data across the two cohorts (Supplementary Figs. 3, 4 and 5 and Supplementary Table 3), with sample quality control done in a GIA-aware manner (Supplementary Fig. 6) (see Methods). For the analyses described below, unless otherwise stated, we restricted the dataset to 29,745 unrelated trio probands from 22 GIA sub-groups (chosen as described in the Methods section 'Sample filtering for the burden analysis') for whom both parents were inferred to come from the same GIA group as the child and from whom at least one parent was inferred to come from the same GIA sub-group as the child (Table 1 and Supplementary Table 2).

### Exome-wide burden analysis

Following our previous work[7], we calculated the expected probabilities of rare (minor allele frequency (MAF) < 0.005; no homozygotes in gnomAD) biallelic genotypes (homozygous non-reference or compound heterozygous genotypes) in each cohort and GIA sub-group separately, taking into account GIA sub-group-specific allele frequencies and autozygosity levels (Supplementary Fig. 7). The four genotype consequence classes we considered included synonymous/synonymous (that is, biallelic synonymous) as a negative control plus three predicted damaging classes: loss-of-function (LoF)/LoF, LoF/functional and functional/functional, in which the 'functional' class included protein-altering variants other than high-confidence LoFs that passed various deleteriousness filters (see Supplementary Note section on 'Filtering of missense and other functional variants').

To quantify the exome-wide recessive burden, we compared the expected number of biallelic genotypes in a given consequence class to the observed number. Extended Data Fig. 3 indicates that these generally agree well for biallelic synonymous genotypes in the largest GIA sub-groups, demonstrating that our quality control is robust. For GIA sub-groups with smaller sample sizes, the observed number of biallelic synonymous genotypes was often significantly lower than expected (Supplementary Table 4), which we suspect is because our estimate of the expected number was inflated owing to overestimation of allele frequencies in the small sample sizes. Therefore, for Fig. 1 and the results reported throughout the text and shown in Fig. 2, Extended Data Figs. 4–7 and Supplementary Figs. 8 and 12, we focused on seven large GIA sub-groups (see Methods section 'Sample filtering for the burden analysis'): AFR4, AMR0, EUR4, EUR5, MDE3, SAS4 and SAS5 (total *n* = 25,523 unrelated probands; Table 1). We used the observed and expected numbers of biallelic genotypes within these different GIA sub-groups to calculate the fraction of patients attributable to autosomal recessive coding causes, which we refer to below as the 'attributable fraction'. In brief, this was calculated as the difference between the observed and expected number of damaging biallelic genotypes divided by the number of probands (Supplementary Table 4 and see Methods section 'Testing for enrichment of biallelic genotypes over expectation'). We calculated the attributable fraction both exome-wide, using all probands in different GIA sub-groups separately or in aggregate (Fig. 1 and Supplementary Figs. 8b and 9), as well as in different subsets of probands and gene sets (Fig. 2 and Extended Data Figs. 4–6).

The estimates of the attributable fraction owing to autosomal recessive coding variants ranged from ~2–7% in AMR0, EUR4, EUR5 and SAS5 to 14.1% (95% CI, 5.7–23.1%) in MDE3 and 18.6% (10.7–26.7%) in SAS4. For all populations, this was lower than the attributable fraction owing to de novo coding mutations (Fig. 1a). The autosomal recessive attributable fraction was significantly correlated with the average level of autozygosity in the GIA groups (*r* = 0.99, *P* = 5 × 10⁻⁶ for the seven GIA sub-groups in Fig. 1b), whereas the attributable fraction owing to de novo mutations was not (*r* = −0.46, *P* = 0.30; Fig. 1b). Thus, despite making up only a small proportion of the total number of probands (6% combined), SAS4 and MDE3 make up 26.0% of the total autosomal recessive attributable fraction across these seven GIA sub-groups (Fig. 1c). Extended Data Fig. 4 shows that the total autosomal recessive attributable fraction is not significantly different between GeneDx and DDD (4.1% versus 3.8%; *P* = 0.23). It was higher in diagnosed than in undiagnosed patients (6.9% versus 2.6%; two-sided test for a difference in proportions, *P* = 3.2 × 10⁻⁶¹) and in females than in males (4.5% versus 3.7%; *P* = 0.001) (Extended Data Fig. 4).

We next examined how much of the exome-wide autosomal recessive attributable fraction was explained by known disease-associated autosomal recessive genes (Extended Data Fig. 5). We considered

**Table 1 | Sample sizes and average autozygosity for the 22 GIA sub-groups included in the analyses after removing probands with cross-continental admixture.** The counts for all GIA sub-groups are shown in Supplementary Table 2. Note that DDD samples from these GIA sub-groups were excluded if there were fewer than 100 unrelated, unaffected parents in DDD

| GIA sub-group | Closest corresponding reference population | Number of unrelated probands | | | Number of unrelated, unaffected parents | | Average $F_{ROH}$ for probands | |
|---|---|---|---|---|---|---|---|---|
| | | DDD | GeneDx | Combined | DDD | GeneDx | DDD | GeneDx |
| AFR3 | West African | – | 96 | 96 | – | 232 | – | 0.0013 |
| AFR4[b] | African-American | – | 843 | 843 | – | 2,109 | – | 0.0003 |
| AMR0[b] | Mexican | – | 1,178 | 1,178 | – | 2,674 | – | 0.0017 |
| AMR3 | Puerto Rican | – | 380 | 380 | – | 894 | – | 0.0027 |
| AMR4 | Latin American[a] | – | 204 | 204 | – | 511 | – | 0.0029 |
| AMR9 | Colombian | – | 260 | 260 | – | 745 | – | 0.0028 |
| EAS1 | East Asian[a] | – | 65 | 65 | – | 205 | – | 0.0003 |
| EAS2 | Chinese | – | 295 | 295 | – | 732 | – | 0.0008 |
| EAS3 | East Asian[a] | – | 128 | 128 | – | 348 | – | 0.0007 |
| EAS5 | Vietnamese/Cambodian | – | 89 | 89 | – | 245 | – | 0.0010 |
| EUR1 | European[a] | – | 657 | 657 | – | 1,694 | – | 0.0021 |
| EUR3 | European[a] | – | 125 | 125 | – | 512 | – | 0.0005 |
| EUR4[b] | Western European | 7,057 | 12,489 | 19,546 | 12,997 | 26,016 | 0.0005 | 0.0004 |
| EUR5[b] | Eastern European | 208 | 1,577 | 1,785 | 459 | 3,909 | 0.0008 | 0.0002 |
| EUR6 | European[a] | – | 38 | 38 | – | 200 | – | 0.0001 |
| EUR7 | Italian | 118 | 1,412 | 1,530 | 294 | 3,620 | 0.0010 | 0.0005 |
| MDE2 | Middle Eastern[a] | – | 79 | 79 | – | 149 | – | 0.0468 |
| MDE3[b] | Middle Eastern[a] | 87 | 571 | 658 | 157 | 1,114 | 0.0351 | 0.0335 |
| MDE4 | Middle Eastern[a] | – | 87 | 87 | – | 167 | – | 0.0347 |
| SAS3 | Bangladeshi | 80 | 109 | 189 | 125 | 242 | 0.0185 | 0.0082 |
| SAS4[b] | Pakistani | 467 | 454 | 921 | 577 | 832 | 0.0539 | 0.0298 |
| SAS5[b] | Indian | 100 | 492 | 592 | 187 | 1,019 | 0.0185 | 0.0108 |
| Total (all GIA sub-groups) | | 8,117 | 21,628 | 29,745 | 14,796 | 48,169 | 0.0044 | 0.0027 |
| Total (seven GIA sub-groups in Fig. 1) | | 7,919 | 17,604 | 25,523 | 14,377 | 37,673 | 0.0043 | 0.0026 |

[a]For some GIA sub-groups, there were no reference samples in the same cluster (Extended Data Fig. 2), so we give only the GIA groups label for these. [b]Indicates the seven GIA sub-groups included in the overall attributable fraction calculations given throughout the text, in Figs. 1 and 2, Extended Data Figs. 4–7 and Supplementary Figs. 8 and 12. $F_{ROH}$, fraction of the genome in runs of homozygosity.

a set of 1,818 known ARDD genes that are used for diagnosis in one or both cohorts, including 1,069 'consensus' genes in both the Developmental Disorders Gene-to-Phenotype Database (DDG2P) list and GeneDx's in-house list, and 749 'discordant' genes in only one of the lists (Supplementary Table 5). Consensus genes explained 68.1% (62.0–74.4%) of the total exome-wide attributable fraction and consensus + discordant genes explained 84.0% (76.9–91.3%) (Fig. 2a). Once the consensus + discordant genes were removed, there was minimal residual burden of damaging biallelic genotypes across the remaining genes (attributable fraction, 0.6% (0.3–1.0%); $P = 0.0003$) (Fig. 2a). Consensus + discordant genes explained 86.9% (78.1–96.1%) of the total exome-wide attributable fraction in probands with European ancestries (EUR4 + EUR5), which was significantly higher than the fraction in those with non-European ancestries (AFR4 + AMR0 + MDE3 + SAS4 + SAS5) (79.8% (67.9–92.3%)); two-sided test for a difference in proportions, $P = 0.003$).

We estimated that 34.4% (27.9–41.4%) of the autosomal recessive attributable fraction in consensus + discordant ARDD genes was explained by variants not annotated as pathogenic or likely pathogenic (P/LP) in ClinVar. The estimate in DDD (46.7% (32.6–62.1%)) was higher than in GeneDx (30.3% (23.1–38.1%)) (Fig. 2b), probably reflecting the fact that GeneDx systematically submits pathogenic variants to ClinVar, whereas DDD does not. This implies that a substantial fraction of the

recessive burden in the GIA groups represented by DDD in particular is a result of variants in known autosomal recessive genes that have not been annotated as P/LP in ClinVar.

We then estimated the autosomal recessive attributable fraction in as-yet-undiagnosed patients ($n_{undiagnosed} = 4{,}425$ and 12,604 for DDD and GeneDx, respectively for the seven GIA groups in Fig. 1) within the set of ARDD genes that were used for diagnosis by the relevant cohort. We estimated that 1.2% (0.7–1.8%) of the as-yet-undiagnosed DDD patients are attributable to damaging biallelic coding variants in autosomal recessive DDG2P genes (Fig. 2c). All of this is a result of biallelic LoF/functional or functional/functional genotypes. Among these 4,425 DDD undiagnosed individuals, 367 damaging biallelic genotypes in autosomal recessive DDG2P genes have been reported back to clinicians through DECIPHER as potentially clinically relevant and have either not yet been clinically evaluated (106 genotypes) or have been classified as being of uncertain significance (261 genotypes). Based on the above attributable fraction estimate, this implies that 15.0% ((1.247% of 4,425) out of 367) of these biallelic genotypes are actually pathogenic. Similarly, in GeneDx, we estimate that 1.6% (1.2–1.9%) of as-yet-undiagnosed patients are attributable to damaging biallelic coding variants in known autosomal recessive disease genes on GeneDx's curated in-house list, of which 87.8% are biallelic LoF/functional or functional/functional (Fig. 2c). Many of these are probably being reported as variants of

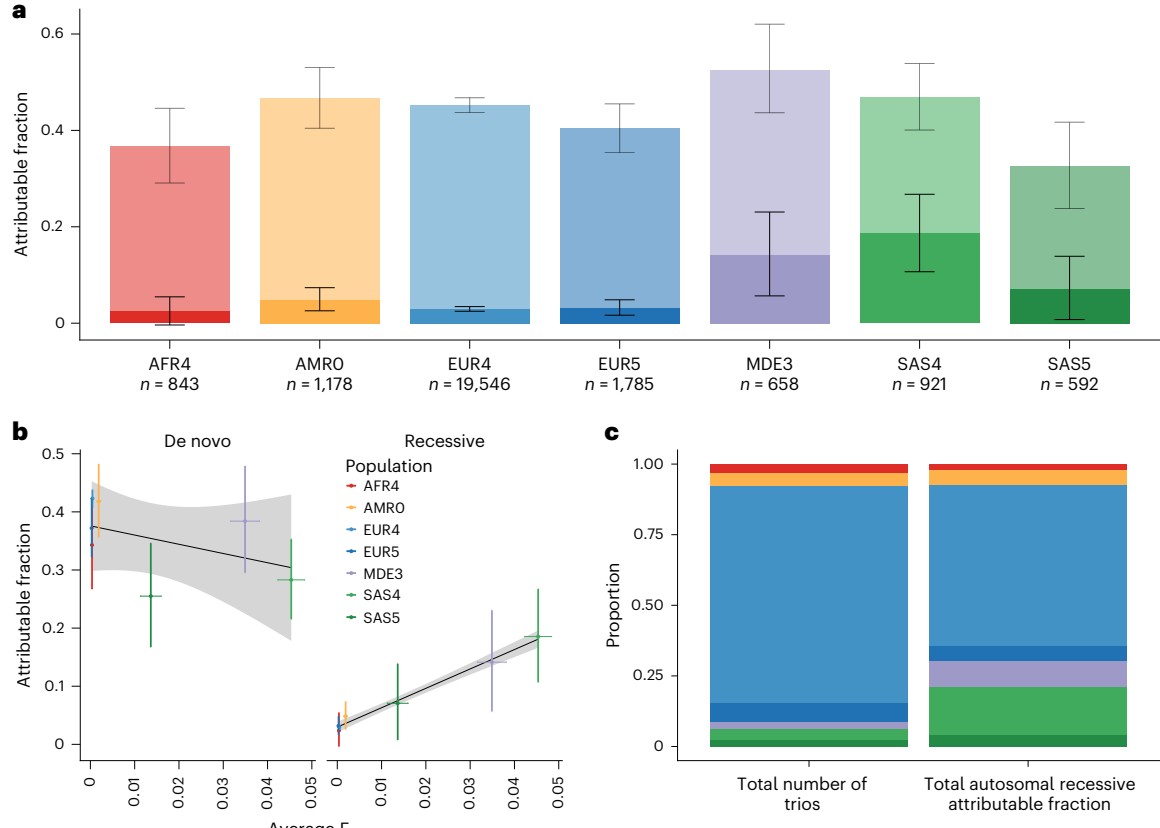

**Fig. 1 | Estimates of the fraction of patients attributable to autosomal recessive coding variants or de novo coding mutations in DDD and GeneDx across seven large GIA sub-groups (n = 25,523). a**, Estimated attributable fraction per GIA sub-group. The de novo attributable fractions (lighter shading) are stacked on the autosomal recessive attributable fractions (darker shading), with the total height of the bars being the sum of the attributable fractions. Lines show 95% confidence intervals (CIs). **b**, Estimated attributable fraction owing to de novo coding mutations (left) or autosomal recessive coding variants (right) versus average autozygosity ($F_{ROH}$) for these seven GIA sub-groups (see Table 1). Colored lines, 95% CIs. The black line is the line of best fit and gray shading shows its 95% CI. **c**, Comparison of the proportion of the total sample size (left) versus the proportion of the total autosomal recessive attributable fraction (right) accounted for by each GIA sub-group.

unknown significance, as they do not currently meet the criteria for being classed as P/LP. This highlights the challenge of interpreting rare missense and other functional variants in ARDD genes.

Among the undiagnosed GeneDx patients, there was a small but significant excess of rare LoF/LoF genotypes in ARDD genes on the diagnostic list, and the attributable fraction of such genotypes was estimated at 0.19% (0.10–0.31%) (Fig. 2c). By contrast, there was no significant burden of LoF/LoF genotypes in autosomal recessive DDG2P genes in undiagnosed DDD patients (attributable fraction, −0.01.% (−0.12–0.18%); $z$-test for a difference in proportions compared to the GeneDx attributable fraction, $P = 0.007$). This may partly reflect the challenge that commercial laboratories face with reanalysis, which requires consent and allocation of resources. Unless reanalysis is ordered by the clinician or a patient is included in a research project, there can be a lag in issuing updated reports when a variant classification is upgraded. By contrast, the iterative reanalysis carried out in a research cohort such as DDD[16,17] ensures that these diagnostic variants in newly defined ARDD genes are identified and reported efficiently.

Finally, we applied exome-wide burden analysis to patients with a single genetic diagnosis to try to quantify the contribution of multi-gene (composite) diagnoses (Supplementary Note). We estimated that up to 12.5% of these patients have an as-yet-unidentified de novo mutation in another gene that also contributes to the phenotype, mostly outside known developmental-disorder-associated genes, and these are enriched among patients whose current diagnosis is deemed 'partial' rather than 'full' (Extended Data Fig. 6). By contrast,

we found that recessive variants do not make a significant contribution to as-yet-undetected composite diagnoses.

## Gene discovery

We next tested for enrichment of damaging biallelic genotypes in each gene to try to identify novel ARDD genes. For the main gene discovery analysis, we included 29,745 unrelated trios without cross-continental admixture from the 22 GIA sub-groups shown in Table 1. We reasoned that although the expected number of biallelic genotypes was being overestimated in some of the smaller GIA sub-groups, this would only reduce our power rather than lead to false positives, and the increased sample size by including these GIA sub-groups might compensate for this. For each gene, we used a Poisson test to compare the total observed number of biallelic genotypes in a given consequence class across GIA sub-groups with the total expected number (Supplementary Fig. 10). We considered four combinations of damaging biallelic genotypes (LoF/LoF, LoF/LoF + LoF/functional, functional/functional and all combined (LoF/LoF + LoF/functional + functional/functional)) and then took the lowest $P$ value per gene. We used a Bonferroni threshold of $P < 7.2 \times 10^{-7}$ (corrected for four tests for each of 17,320 genes).

We found 24 genes that passed Bonferroni correction and an additional 42 genes that passed a false discovery rate (FDR) of 5% (Supplementary Data File). Twenty-two of the Bonferroni significant genes (22 out of 24 = 92%) and 61 of the FDR < 5% genes (61 out of 66 = 92%) are known ARDD genes on the GeneDx and/or DDG2P lists. Table 2 shows the five genes passing FDR < 5% that are not ARDD

genes on either list: *CRELD1*, *KBTBD2*, *ZDHHC16*, *HECTD4* and *ATAD2B*. Of these, *CRELD1* and *KBTBD2* passed Bonferroni correction, and *CRELD1*[18] and *HECTD4*[19] were recently reported. After repeating the gene-based tests using only undiagnosed probands ($n = 17{,}029$ trios), *ZDHHC16* also passed Bonferroni correction ($P = 6.05 \times 10^{-7}$ versus original $P = 3.04 \times 10^{-6}$) (Table 2 and Supplementary Data File). Using semantic similarity scores[20] of HPO terms between pairs of patients, we found that the patients with damaging biallelic genotypes in the consensus + discordant genes and in the five novel FDR < 5% genes were significantly more phenotypically similar to each other than were randomly chosen patients (one-sided Wilcoxon rank sum, $P = 2.2 \times 10^{-126}$ and $P = 0.0058$, respectively; Extended Data Fig. 7). In Supplementary Table 6, we present the observed deleterious biallelic variants in these five genes from both patients included in the discovery analysis as well as additional patients identified subsequently, together with the patients' associated HPO terms. Details of the patients with damaging biallelic genotypes in *CRELD1, ZDHHC16, HECTD4* and *ATAD2B* are given in the Supplementary Note.

We observed two patients with damaging LoF/functional compound heterozygous genotypes in *KBTBD2* ($P = 1.3 \times 10^{-7}$). This gene encodes Kelch repeat and BTB domain-containing protein 2, which is an adaptor of a ubiquitin ligase complex that regulates insulin signaling[21]. Specifically, one of its downstream targets is p85α, the regulatory subunit of phosphoinositol-3-kinase, which drives a key pathway on which insulin signaling depends[21]. *KBTBD2* is highly expressed in mouse brain as well as adipose tissue, liver and muscle, and knocking it down in mice resulted in elevated expression of p85α in all of these tissues and a phenotype involving lipodystrophy, hepatic steatosis, insulin resistance, severe diabetes and growth retardation[21]. Consistent with this finding, both of our patients displayed some degree of growth retardation. The older patient (who was recruited during adolescence) had hyperglycemia and diabetes; the younger patient was below the age at which diabetes might be expected to develop (Supplementary Table 6). Thus, the phenotypes in these two patients appear to be consistent with the mouse knockout but with some additional phenotypic features (microcephaly, cardiomyopathy, developmental delay). Consistent with the neurodevelopmental phenotypes we observe, *KBTBD2* is part of a co-expression module enriched in human fetal brain[22]. We subsequently identified a patient from CENTOGENE with a homozygous LoF (p.(Val433fs)) in this gene, whose phenotype included intrauterine growth retardation, microcephaly and dysmorphic features. She had no documented hyperglycemia or features suggestive of diabetes, but she died at the age of 3 months, before the age at which this might be expected to develop. More details about this patient and two similarly affected siblings are given in the Supplementary Note.

We repeated the gene-based tests after applying a stricter filter for admixed probands (requiring both parents to come from the same GIA sub-group as the child; $n = 23{,}574$ trios) and after removing the filter of probands with inferred cross-continental admixture ($n = 32{,}058$ trios). No additional novel genes were identified and key conclusions were unchanged (Supplementary Data File).

## Discussion

We have examined the contribution of autosomal recessive coding variants to developmental disorders in the largest sample to date, containing about six times more trios and greater ancestral diversity than our previous work in an earlier release of the DDD study[7]. The current study demonstrates the power of federated analysis of

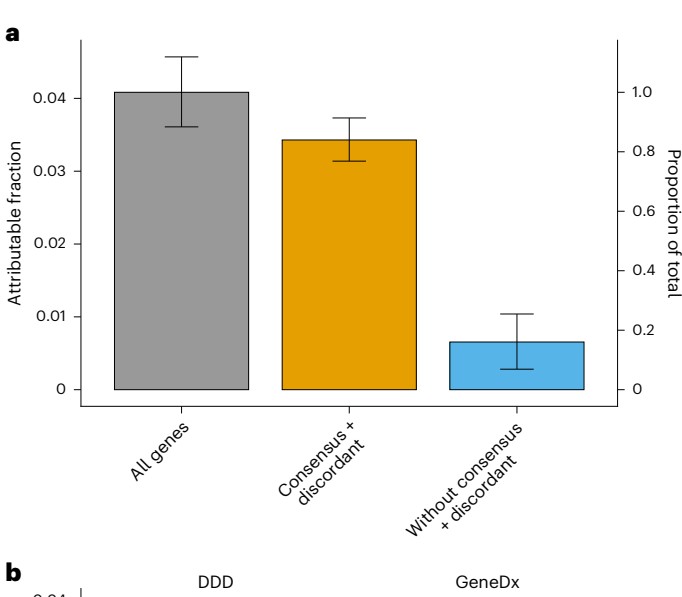

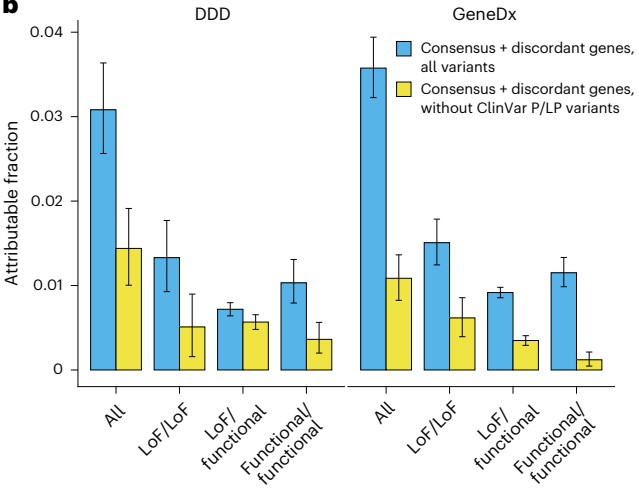

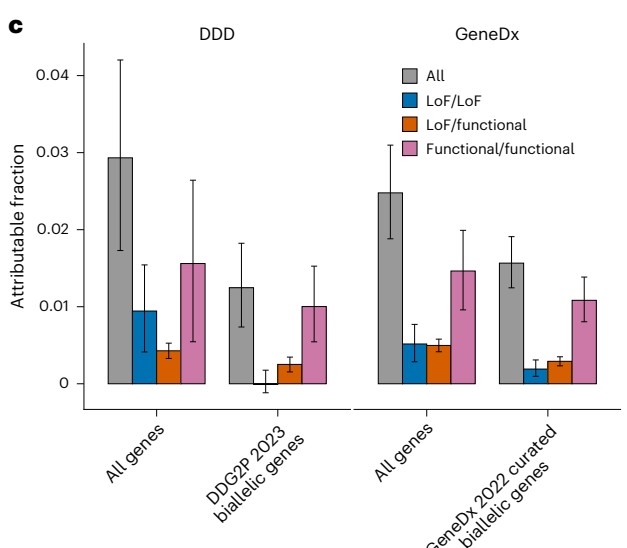

**Fig. 2 | Estimates of the fraction of patients attributable to autosomal recessive coding variants in different subsets of genes and patients.** These plots are focused on the individuals without cross-continental admixture from seven large GIA sub-groups, as in Fig. 1 and Table 1. **a**, Estimates in all individuals from DDD and GeneDx combined ($n = 25{,}523$), for all genes versus genes in the indicated lists. **b**, Estimates in all individuals for consensus + discordant genes split by cohort ($n = 7{,}919$ and $17{,}604$ for DDD and GeneDx, respectively), comparing the estimates obtained with all variants versus after removing variants annotated as pathogenic or likely pathogenic (P/LP) in ClinVar. **c**, Estimates in undiagnosed individuals ($n = 4{,}425$ and $12{,}604$ for DDD and GeneDx, respectively), for all genes versus the genes that are used for clinical filtering of diagnostic autosomal recessive variants in the respective cohorts, split by cohort and functional consequence of the variants. Error bars, 95% confidence intervals.

**Table 2 | Genes not in the consensus or discordant lists that passed FDR < 5% in the main gene burden analysis, which was based on 29,745 probands without inferred cross-continental admixture (referred to as 'all individuals' in the column header).** *P* values of <7.2×10⁻⁷ pass Bonferroni correction. We only show the result for the most significant combination of consequence classes per gene. Additionally, we show the *P* value obtained by restricting the analysis to 17,029 undiagnosed probands without inferred cross-continental admixture. Both sets of *P* values are from a one-sided Poisson test. The Supplementary Data File shows results for all combinations of consequence classes. The Supplementary Note gives more detail on these genes, including on the putative alternative partial diagnosis in one of the *ATAD2B* cases that led to this individual being dropped from the analysis of undiagnosed cases

| HGNC symbol | Most significant variant class | Results from all individuals | | | Supporting evidence and notes |
|---|---|---|---|---|---|
| | | Observed (expected) | *P* value | *P* value in undiagnosed only | |
| *CRELD1* | All | 6 (0.2067) | $9.08 \times 10^{-8}$ | $8.83 \times 10^{-9}$ | Recently implicated gene[18]<br>Cosegregation in one affected sibling<br>Observed in an additional proband who was removed owing to admixture, plus two additional probands in a newer GeneDx datafreeze<br>Known dominant developmental disorder gene |
| *KBTBD2* | LoF/functional | 2 (0.0005) | $1.25 \times 10^{-7}$ | $6.77 \times 10^{-8}$ | Similar phenotype in mouse model[21,27]<br>Additional case with similar phenotype identified in CENTOGENE<br>Extremely constrained (pLI=1) |
| *ZDHHC16* | LoF/LoF + LoF/functional | 3 (0.0265) | $3.04 \times 10^{-6}$ | $6.05 \times 10^{-7}$ | Cosegregation in one affected sibling<br>Three additional cases with similar phenotypes identified (one from CENTOGENE, two from GeneDx)<br>Zebrafish model shows defective telencephalon development[28] |
| *HECTD4* | LoF/LoF | 2 (0.0042) | $8.80 \times 10^{-6}$ | $2.79 \times 10^{-6}$ | Recently implicated gene[19]<br>Two additional cases with similar phenotypes identified in GeneDx<br>Extremely constrained (pLI=1) |
| *ATAD2B* | LoF/LoF + LoF/functional | 2 (0.0045) | $1.02 \times 10^{-5}$ | $2.81 \times 10^{-3}$ | Cosegregation in one affected sibling<br>Extremely constrained (pLI=1)<br>Homozygous mouse knockout shows behavioral abnormalities[29]<br>One additional case with damaging biallelic missense variants identified in GeneDx<br>Contrary evidence: limited phenotypic similarity between patients other than developmental delay or intellectual disability |

pLI, probability of being LoF-intolerant.

large multi-ancestry cohorts from which genome-wide individual-level data cannot be combined in a single location owing to data governance considerations. Transferring deidentified summary data (principal components and their loadings) between cohorts allowed us to identify individuals with similar GIA across cohorts (Extended Data Fig. 2), boosting power, particularly for smaller and historically understudied groups. Using these multiple GIA groups, we showed that the average level of autozygosity is a strong predictor of the fraction of patients in a given sample who are attributable to autosomal recessive causes (Fig. 1b and Supplementary Fig. 9).

We found that the majority of the autosomal recessive burden is explained by known ARDD genes and that this is true both for European-ancestry and non-European-ancestry individuals (87% versus 80%). By contrast, our 2018 paper[7] reported that among DDD probands with European and Pakistani GIA (roughly corresponding to the EUR4 and SAS4 GIA sub-groups in this work), the autosomal recessive DDG2P genes known at the time explained 48% of the recessive burden. Extended Data Fig. 5 shows that the most recent DDG2P ARDD gene list explains a higher fraction of the burden in this analysis than the list used in our previous paper[7] and that the GeneDx ARDD gene list explains the highest fraction. This suggests that DDG2P is a more conservatively curated list than the GeneDx list. It also reflects the success of worldwide autosomal recessive gene discovery efforts over the last 6 years, many of which have been conducted through MatchMaker Exchange-style approaches[14]. The comparably modest yield of new ARDD genes in this work suggests that this kind of statistical analysis of large, relatively unselected cohorts may not be the most efficient way to find new autosomal recessive genes. A more efficient approach may entail identifying new candidate genes in unsolved cases from geographically isolated populations or consanguineous families with multiple affected individuals and then finding additional

cases through Matchmaker Exchange, DECIPHER or other global data-sharing initiatives.

Our paper contains several clinically important messages. Firstly, the fact that most of the autosomal recessive burden is in known ARDD genes suggests that if a patient undergoes sequencing and is not found to have any candidate putatively damaging biallelic genotypes in these genes (of which only a subset may meet the American College of Medical Genetics and Genomics (ACMG) P/LP criteria), their residual risk of having an autosomal recessive condition, at least as a result of coding variants, is low. However, it does depend on their degree of consanguinity; from the attributable fraction estimates in our dataset, we estimate this residual risk at 0.39% (0.14–0.65%) if the patient has a fraction of the genome in runs of homozygosity ($F_{ROH}$) of <0.0156 (the expectation for offspring of second cousins) and 6.3% (−0.4–13.4%) otherwise ($F_{ROH}$ > 0.0156). Secondly, our estimates suggest that a substantial fraction of potential diagnoses in known autosomal recessive genes are being missed, mostly those involving missense variants, which remain challenging to interpret (Fig. 2c). For example, among the 7,732 DDD individuals included in our main burden analysis, there are 230 confirmed autosomal recessive diagnoses in DDG2P genes, and the attributable fraction estimate in Fig. 2c suggests that there are an additional ~73 diagnoses to be made in these genes among the 4,425 undiagnosed patients (1.65% of 4,425); thus, we are missing ~24.1% of diagnoses (73 out of 303) in autosomal recessive DDG2P genes. Thirdly, our results also imply that if we could find all the possible diagnoses in established ARDD genes by better distinguishing pathogenic from benign functional variants, we would probably diagnose about twice as many patients as we would by discovering new ARDD genes, at least in the GIA groups under study here. Our attributable fraction estimates suggest that among undiagnosed DDD and GeneDx patients (17,029), there are ~280 diagnoses yet to be made from LoF/functional and

functional/functional genotypes in the ARDD genes that are already used for reporting by the relevant cohort versus ~161 diagnoses to be made from as-yet-undiscovered ARDD genes. Fourthly, the fact that ~47% of the burden in established ARDD genes in DDD is not explained by ClinVar P/LP variants suggests that recessive carrier screening, which currently tends to focus on such variants[23], has the potential to be extended in the future as knowledge about which variants are clinically significant increases. Finally, our burden analyses conducted in patients who already have a single genetic diagnosis imply that in as many as ~12.5% of these patients (~743), an as-yet-unidentified de novo mutation in another gene also contributes to the phenotype; almost all of this burden is outside known monoallelic and X-linked dominant DDG2P genes (Supplementary Note and Extended Data Fig. 6). If these contributing de novo mutations could be identified, it would more than double the number of patients in these cohorts who currently have a composite genetic diagnosis (*n* = 596). We also find that these as-yet-undetected composite diagnoses are more likely among patients whose current single diagnosis is deemed 'partial' than in those in whom it is deemed 'full' (attributable fraction 18% versus 5.7%) and that recessive variants are unlikely to contribute to further composite diagnoses (Supplementary Note). The 12.5% estimate is higher than previous estimates of the rate of composite diagnoses[24], and this may be for several reasons. Firstly, the excess burden in patients who currently have a single diagnosis may not only reflect dual diagnoses but may also partly reflect digenic or oligogenic causes, whereby the second variant may have a role but be insufficient on its own to cause disease. Secondly, as noted below, it may reflect ascertainment bias in the DDD and GeneDx cohorts.

Although the vast majority of the autosomal recessive burden was explained by known genes in the GeneDx and/or DDG2P lists, we did identify several new or only recently described genes with compelling or suggestive evidence for causation (Table 2). Overall, we believe there is strong evidence that *CRELD1*, *KBTBD2*, *ZDHHC16* and *HECTD4* are bona fide ARDD genes, whereas the current evidence for *ATAD2B* is more equivocal (Supplementary Note). Of the 15 damaging biallelic genotypes contributing to the discovery of these five genes, 11 were LoF/functional or functional/functional. Our stringent missense filtering (Supplementary Figs. 8 and 12) boosted our power to implicate these genes; had we done more lenient missense filtering, the genes highlighted in Table 2 would all have had less significant *P* values and only *KBTBD2* would have passed Bonferroni correction (Supplementary Fig. 12).

This work has several limitations. Firstly, the families studied are not a random sample of the developmental disorder patient population and may be depleted of easy-to-solve families with recessive conditions. Thus, we may have underestimated the contribution of autosomal recessive variants to developmental disorders as a whole, overestimated the true rate of composite diagnoses or overestimated the overall fraction of new diagnoses that could be made by better interpreting missense variants in known ARDD genes. The ascertainment bias also probably explains why hundreds of known ARDD genes did not reach formal Bonferonni significance in our study; we emphasize that the genes that passed Bonferroni correction in this work are not the only bona fide ARDD genes. Secondly, our estimates of attributable fraction assume that every excess damaging biallelic genotype over expectation fully 'explains' one proband (that is, fully penetrant monogenic causes), which may over-simplify the genetic architecture. Thirdly, although our sample contained considerable ancestral diversity, it is clearly not representative of the global population, and the sample sizes for many GIA sub-groups were too small to obtain precise estimates of the attributable fraction. This also undoubtedly reduced our power for gene discovery, given that parental allele frequencies are overestimated in these small samples. Fourthly, we restricted our analyses to variants that fall within the intersection of the bait regions of the eight different exome capture kits used across both cohorts,

which may have led us to underestimate the total attributable fraction. Finally, we focused on protein-coding single-nucleotide variants and small indels, so our estimates do not include the recessive contribution of noncoding variants or copy number variants.

In conclusion, the discovery of the remaining ARDD genes will require larger samples and/or more focused sampling of genetically isolated communities enriched for causal founder variants and/or consanguineous families with multiple affected individuals. However, these as-yet-undiscovered genes are unlikely to account for a high fraction of neurodevelopmental disorder patients, at least in the GIA groups represented in this study. To maximize diagnostic yield, future work should develop better strategies to distinguish pathogenic from benign recessive functional variants, such as approaches involving multiplex assays of variant effects[25,26].

## Online content

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

[1]Wellcome Sanger Institute, Wellcome Genome Campus, Hinxton, UK. [2]Department of Clinical and Biomedical Sciences, University of Exeter Medical School, Royal Devon and Exeter Hospital, Exeter, UK. [3]GeneDx, Gaithersburg, MD, USA. [4]Deka Biosciences, Germantown, MD, USA. [5]MRC Epidemiology Unit, Cambridge, UK. [6]Institute of Developmental and Regenerative Medicine, Department of Paediatrics, University of Oxford, Oxford, UK. [7]Geisinger, Danville, PA, USA. [8]GeneDx Iceland, Reykjavík, Iceland. [9]Istanbul Medipol University, Medical School, Department of Medical Genetics, Istanbul, Turkey. [10]Marmara University Medical Faculty, Pendik Training and Research Hospital, Department of Pediatric Neurology, Istanbul, Turkey. [11]The Royal Hospital, Muscat Al Ghubra Area 111 Seeb, Muscat, Oman. [12]Medical Genetics, CENTOGENE GmbH, Rostock, Germany. [13]Clinic of Internal Medicine, Department of Hematology, Oncology, and Palliative Medicine, University Medicine Rostock, Rostock, Germany. [14]Laboratory of Human Genetics & Therapeutics, BESE, KAUST, Thuwal, Saudi Arabia. [15]Leeds Institute of Medical Research, University of Leeds, St. James's University Hospital, Leeds, UK. [16]Yorkshire Regional Genetics Service, Chapel Allerton Hospital, Leeds, UK. [17]Cambridge University Hospitals Foundation Trust, Addenbrooke's Hospital, Cambridge, UK. [18]Program in Medical and Population Genetics, Broad Institute of MIT and Harvard, Cambridge, MA, USA. [19]Analytic and Translational Genetics Unit, Massachusetts General Hospital, Boston, MA, USA. [20]Center for Genomic Medicine, Massachusetts General Hospital, Boston, MA, USA. [21]These authors contributed equally: Vincent D. Ustach, Hilary C. Martin. ✉e-mail: hcm@sanger.ac.uk

## Methods

### Cohorts, sequencing, alignment and variant calling

**DDD.** Between April 2011 and April 2015, the DDD study recruited a total of 13,450 patients (88% in a trio) with a severe developmental disorder who remained undiagnosed after undergoing the typical clinical genetics investigations[30]. The phenotypic inclusion criteria included neurodevelopmental disorders, congenital abnormalities, growth abnormalities, dysmorphic features and unusual behavioral phenotypes. Recruitment took place across 24 regional genetics services within the United Kingdom and the Republic of Ireland health services. The families gave their informed consent to participate, and the study was approved by the UK Research Ethics Committee (10/H0305/83, granted by the Cambridge South Research Ethics Committee; GEN/284/12, granted by the Republic of Ireland Research Ethics Committee).

Details on sample collection, exome sequencing, alignment, variant calling and variant annotation have been described previously[7,8,10]. In brief, exome capture was carried out with either the Agilent Sure-Select Human All Exon V3 or V5 baits. We used the Burrows–Wheeler Aligner (BWA) aln algorithm (BWA v.0.5.10) and the BWA mem algorithm (BWA v.0.7.12)[31,32] to align reads to the GRCh37 1000 Genomes Project phase 2 reference (hs37d5). Picard Markduplicates (v.1.98 and v.1.114; https://broadinstitute.github.io/picard) and Genome Analysis Toolkit IndelRealigner (GATK v.3.1.1 and v.3.5.0)[33–35] were used for sample-level binary alignment map (BAM) improvement. To call single-nucleotide variants and indels, we used the GATK HaplotypeCaller, CombineGVCFs and GenotypeGVCFs (GATK v.3.5.0). Variant calling was restricted to the merged bait regions from the Agilent V3 and V5 exome capture kits used in the sequencing plus a padding region of 100 bp on either side.

**GeneDx.** Patients were referred to GeneDx for clinical whole-exome sequencing for the diagnosis of suspected Mendelian disorders[6,36]. Patient medical records were converted into HPO terms using Neji concept recognition[37], with manual review by laboratory genetic counselors or clinicians. Patients were selected for inclusion in this study based on having one or more HPO terms from a list of 716 that fell under 'abnormality of the nervous system'[30]. The study was conducted in accordance with all guidelines set forth by the Western Institutional Review Board (WIRB 20162523). Informed consent for genetic testing was obtained from all individuals undergoing testing, and the Western Institutional Review Board waived authorization for the use of deidentified aggregate data. Individuals or institutions who opted out of this type of data use were excluded.

The samples underwent exome sequencing[36] with a version of either SureSelect Human All Exon V4 (Agilent Technologies), Clinical Research Exome (Agilent Technologies) or xGen Exome Research Panel v.1.0 (IDT), or genome sequencing with the Kappa HyperPrep PCR-Free kit. They were sequenced with either 2 × 100 bp or 2 × 150 bp reads on HiSeq 2000, 2500, 4000 or NovaSeq 6000 (Illumina). Reads were mapped to the published human genome build UCSC hg19/GRCh37 reference sequence using the BWA (using v.0.5.8 to v.0.7.8, depending on the time of sequencing)[31,32]. BAM files were then converted to CRAM format with Samtools (v.1.3.1)[38] and indexed. Individual gVCF files were called with GATK v.3.7-0 HaplotypeCaller[33–35] in GVCF mode by restricting output regions to the RefGene primary coding regions ±50 bp. Single-sample gVCF files were then combined into multi-sample gVCF files, with each combined file containing 200 samples. These multi-sample GVCF files were then jointly genotyped using GATK v.3.7-0 GenotypeGVCFs. GATK v.3.7-0 VariantRecalibrator (VQSR) was applied for both single-nucleotide polymorphisms and indels, with known single-nucleotide polymorphisms from 1000 Genomes phase 1 high confidence set and 'gold standard' indels from a previous publication[39].

**CENTOGENE.** Following the main analysis of DDD and GeneDx, we queried the CENTOGENE Biodatabank to try to identify additional individuals with potentially diagnostic biallelic variants in our new recessive genes. The CENTOGENE Biodatabank holds data from nearly 900,000 individuals from over 120 countries, over 70% of whom are of non-European descent. From these, nearly 412,000 are affected with a wide range of rare and neurodegenerative diseases.

### Ancestry assignment

**Assigning broad-scale GIA groups.** To identify samples with similar genetic ancestry, we subset the genotypes of samples from 1000 Genomes phase 3 (ref. 40) and the Human Genome Diversity Project (HGDP)[41] to the common single-nucleotide variants (MAF > 0.01) with low missingness (<10%). We first removed any related samples (see Supplementary Note) and then ran pairwise linkage disequilibrium pruning using plink[42] (--indep-pairwise 50 5 0.2). We applied hard genotype filters to the DDD and GeneDx datasets (GQ > 20, DP > 7 and, subsequent to this, genotype missingness of <10%), and took the intersection of the remaining variants across the datasets ($n = 17,693$ single-nucleotide variants). The first 20 principal components (PCs) of the 1000 Genomes and HGDP reference cohorts were calculated using GCTA (v.1.93.0)[43,44]. Using the single-nucleotide polymorphism loadings, we projected the DDD and GeneDx samples onto the reference sample PCs. Using the first seven PCs, we ran uniform manifold approximation and projection (UMAP)[45] with the umap-learn Python package, with parameters - min-dist=0 and n_neighbours=100. This created the six clusters seen in Extended Data Fig. 1, with the corresponding labels applied to DDD and GeneDx samples based on the locations of the reference individuals with known ancestry. We refer to these as the GIA groups.

**Fine-scale ancestry.** To assign fine-scale genetic ancestry to the individuals within each GIA group, we ran PCA on the individuals from 1000 Genomes and HGDP as well as the unrelated parents in the GeneDx dataset from that GIA group. We then projected the remaining samples from GeneDx and the DDD samples from that GIA group onto these PCs. To assign individuals to GIA sub-groups, we took the PCs that captured the majority of the variation in the broad-scale GIA group within which they fell, used these as an input for UMAP and then ran HDBSCAN (a clustering algorithm)[46] on the UMAP coordinates to create the fine-scale clusters that we refer to as GIA sub-groups (Extended Data Fig. 2).

### Runs of homozygosity

To call the runs of homozygosity, we used bcftools-roh[47]. See Supplementary Note for further details.

### Variant, genotype and sample quality control

Variant-level, genotype-level and sample-level quality control are detailed in the Supplementary Note.

### Variant filtering and annotation

**Initial variant filtering.** We calculated the allele frequencies among unrelated, unaffected parents within all GIA sub-groups in our data who have ≥ 150 unrelated, unaffected parents from the two cohorts separately. We retained variants with MAF ≤ 0.005 within all GIA sub-groups available and with MAF < 0.005 in all gnomAD v2.1.1 GIA groups[48] and removed any that had any homozygous genotypes in gnomAD.

We removed any variants that overlap with a known recent segmental duplication[49] or a simple tandem repeat[50] obtained from the UCSC browser[51]. We also removed any variants that do not overlap the intersection of the bait regions from all exome captures used in the sequencing of both cohorts.

**Variant annotation.** Both cohorts were annotated using VEP v.94.5 (ref. 52) including the LOFTEE plugin[48]. We focused on the annotation in the canonical transcript to group variants into three classes: synonymous, LoF and functional. Synonymous variants with a maximum

SpliceAI[53] score of <0.1 were used as a control. Our classifications of LoF and functional variants were adapted from those used in gnomAD[48] (https://github.com/broadinstitute/gnomad_methods/blob/main/gnomad/utils/vep.py). We classified the following VEP-predicted consequences as LoF, including only those that were predicted as high-confidence LoFs by LOFTEE: splice_donor_variant, splice_acceptor_variant, stop_gained, frameshift_variant, stop_lost and transcript_ablation. We grouped the following predicted consequences into a group we call 'functional variants': missense_variant, inframe_insertion, inframe_deletion, start_lost, transcript_amplification, protein_altering_variant, splice_region_variant, LoFs predicted to be low confidence by LOFTEE and synonymous variants with a minimum SpliceAI score of 0.8. We removed variants if the predicted consequence in the canonical transcript did not fit into any of the categories listed above.

**Filtering of missense and other functional variants.** We explored the use of different metrics for predicting the deleteriousness of missense variants, as described in the Supplementary Note. See also Supplementary Figs. 8, 11 and 12.

**De novo mutations.** The quality control and filtering of the de novo mutations and details of the analysis are given in the Supplementary Note.

## Burden analysis
**Sample filtering for the burden analysis.** We removed trios in which both parents were inferred to come from different GIA sub-groups to the proband. Unless stated otherwise, we also removed trios with cross-continental admixture; that is, in which one parent was inferred to come from a different GIA group to the proband. The Supplementary Data File and Supplementary Table 4 also include results from sensitivity analyses in which we either removed all probands with any admixture ('strict admixture filtering'; that is, in which either parent was inferred to come from a different GIA sub-group from the child) or did no additional admixture filtering other than requiring at least one parent to come from the same GIA sub-group as the child. We restricted all analyses to the 22 GIA sub-groups listed in Table 1, which were those with at least 150 unrelated, unaffected parents and a proportion of probands with cross-continental admixture <0.15. Figures 1 and 2, Extended Data Figs. 4–6 and Supplementary Fig. 8 show exome-wide burden results for seven large GIA sub-groups combined (AFR4, AMR0, EUR4, EUR5, MDE3, SAS4 and SAS5), which were those with at least 500 trios for which the observed number of biallelic synonymous genotypes did not differ significantly from expectation (see below) and for which the exome-wide burden estimates were consistent when carrying out strict admixture filtering (Supplementary Table 4).

**Calculating the observed and expected number of biallelic genotypes.** We extracted all observed trio genotypes with a rare allele in the proband or either parent to calculate the observed and expected number of biallelic genotypes. We removed genotypes within a trio if there was a Mendelian error or if any one of the three individuals had a missing genotype.

The expected number of biallelic genotypes per person was calculated in the same way as previously described[7] and summarized here. We considered four classes of biallelic genotype: LoF/LoF, LoF/functional, functional/functional and synonymous/synonymous. In short, the expected number of biallelic genotypes per person in GIA sub-group $p$, in variant class $c$, in gene $g$ was calculated as:

$$E(b_{c,p,g}) = N_{probands,p}\lambda_{c,p,g}$$

where $N_{probands,p}$ is the number of unrelated probands in GIA sub-group $p$ and $\lambda_{c,p,g}$ is the expected frequency of biallelic genotypes given by

$$\lambda_{c,p,g} = (1 - a_{p,g})f_{c,p,g}^2 + a_{p,g}f_{c,p,g}$$

where $f_{c,p,g}$ is the cumulative frequency of parental haplotypes containing at least one variant of class $c$ in gene $g$ with MAF < 0.005 in GIA sub-group $p$ and $a_{p,g}$ is the proportion of probands in GIA sub-group $p$ with a ROH overlapping gene $g$. For the case of LoF/functional compound heterozygous genotypes, the expected frequency was calculated as:

$$\lambda_{LoF/functional,p,g} = (1 - a_{p,g})\left[2f_{LoF,p,g}f_{functional,p,g}(1-f_{LoF,p,g})\right]$$

To calculate the cumulative frequency, we counted the number of haplotypes with at least one variant of class $c$ in gene $g$ in GIA sub-group $p$ among unrelated, unaffected parents ($h_{c,p,g}$) and divided this by the total number of haplotypes in that group ($N_{haps}$):

$$f_{c,p,g} = h_{c,p,g}/N_{haps}$$

To calculate $h_{c,p,g}$, within a gene, variant class and GIA sub-group for each unrelated, unaffected parent, we counted two haplotypes if a homozygous alternative genotype was observed and one haplotype if a single heterozygous genotype was observed. When a parent had multiple heterozygous variants in that class within the gene, we tried to infer their phase based on transmission to the child, then counted one haplotype if they were in *cis* or two otherwise. If the phase was not clear, we counted one haplotype.

We determined compound heterozygous genotypes as those for which the proband inherited at least one heterozygous variant in the relevant class from each parent within a gene. To determine the observed count of biallelic genotypes, we counted the number of individuals with at least one homozygous alternative or compound heterozygous genotype within a variant class in the gene. If multiple deleterious biallelic genotypes were observed in a given individual, we only counted the one with the most severe consequence (for example, if an individual had both a LoF/LoF and a LoF/functional compound heterozygous genotype in the same gene, this was counted as only LoF/LoF).

**Testing for enrichment of biallelic genotypes over expectation.** To determine the exome-wide burden of biallelic genotypes in variant class $c$ in GIA sub-group $p$, we summed the observed and expected number across genes (that is, $O_{c,p} = \sum_g O_{c,p,g}$ and $E_{c,p} = \sum_g E_{c,p,g}$) and compared these using a Poisson test. For the deleterious classes (LoF/LoF, LoF/functional and functional/functional), we used a one-sided Poisson test to determine whether the observed number was significantly greater than expected, whereas for the biallelic synonymous class, we used a two-sided test. To determine the fraction of cases attributable to damaging biallelic genotypes, we calculated $O_{c,p}$ and $E_{c,p}$ for the three deleterious classes, then calculated the attributable fraction for GIA sub-group $p$ as $(\sum_c O_{c,p} - \sum_c E_{c,p})/N_p$, where $N_p$ is the number of unrelated probands in $p$. Within each GIA sub-group in each cohort, we chose the linkage disequilibrium thinning threshold for calling ROHs (see Supplementary Note) that gives us the maximum $P$ value in the synonymous variant class (given in Supplementary Table 4), then used this linkage disequilibrium thinning threshold to calculate $E_{c,p,g}$ for all variant classes for that GIA sub-group. We calculated the observed and expected values in DDD and GeneDx individually and also conducted a pooled analysis by summing $O_{c,p,g}$ and $E_{c,p,g}$ across cohorts (Supplementary Table 4). The pooled estimates of attributable fraction calculated across the three deleterious genotype classes and across several GIA sub-groups (Fig. 1 and Extended Data Fig. 4) were calculated as: $\sum_p \left[(\sum_c O_{c,p} - \sum_c E_{c,p})\right]/\sum_p N_p$.

For our exome-wide burden analyses (Fig. 1, Supplementary Table 4, Extended Data Figs. 3–6 and Supplementary Figs. 8 and 9), we removed genes flagged in gnomAD v.2 as having an outlying number of synonymous variants, too many missense variants or too many LoF variants[48] along with genes that do not overlap with the intersection

of the bait regions across all the exome capture kits used in DDD or GeneDx. This left 17,320 genes, of which 16,424 had at least one variant that passed our filtering.

In the Discussion, we present estimates of the residual risk of having an autosomal recessive condition for undiagnosed patients without any candidate putatively damaging variants in known ARDD genes. To estimate this factor, we first removed individuals considered diagnosed and individuals with a damaging biallelic genotype who passed our filtering in a consensus or discordant gene. We then split the rest of the individuals into high ($F_{ROH} > 0.0156$) and low ($F_{ROH} < 0.0156$) autozygosity groups and, within each of these groups, calculated the attributable fraction in the genes not on the consensus or discordant gene lists.

**Burden explained by ClinVar pathogenic variants.** See the Supplementary Note for details of how pathogenic ClinVar variants were defined.

**Per-gene tests and multiple testing correction.** For the per-gene enrichment tests, we initially tried implementing the original previously published method[9], which is the exact probability for a sum of independent binomials. However, this method involves calculating all the possible ways that the observed biallelic genotypes could have been distributed across the GIA sub-groups, and this proved to be computationally intractable for genes with high counts given our large sample size. Thus, we instead treated the total count of biallelic genotypes across GIA sub-groups as a sum of Poisson-distributed random variables with rates $\lambda_1, \lambda_2, \cdots, \lambda_n$. This value follows a Poisson distribution with rate $\lambda_1 + \lambda_2 + \cdots + \lambda_n$. Thus, we summed the observed and expected values across GIA sub-groups for a given gene and ran a one-sided Poisson test to determine the probability of observing at least $\sum_p O_{c,p,g}$ genotypes given the expected number, $\sum_p E_{c,p,g}$. Supplementary Fig. 10 shows that this sum-of-Poissons approach gives similar $P$ values to the previous approach (Pearson correlation, $R = 0.98$), particularly for genes with $P < 0.05$, which are the ones of interest.

On each gene, we conducted four non-independent tests for these four probable damaging classes of variant:

- LoF/LoF
- LoF/LoF + LoF/functional
- Functional/functional
- LoF/LoF + LoF/functional + functional/functional

As a Bonferroni correction, we used $P < 0.05/(17320 \times 4) = 7.2 \times 10^{-7}$. We used estimates in all four damaging classes to calculate the Benjamini–Hochberg FDR-adjusted $P$ values. We also implemented a test based on synonymous/synonymous genotypes as a control and, reassuringly, the $P$ values from this calculation followed the expected null distribution (Supplementary Fig. 10c).

**Residual burden in diagnosed individuals.** To estimate the residual burden of de novo and recessive damaging genotypes in diagnosed individuals (Extended Data Fig. 6), we took all individuals with a variant confirmed as diagnostic ('category 1') in the GeneDx cohort along with those defined as diagnosed in DDD according to clinical assertion or autocoded ACMG prediction[4] who overlap our seven GIA groups and who do not have a known or predicted composite diagnosis. We then removed the diagnostic variant from our analyses and repeated the calculation of the attributable fraction for damaging biallelic genotypes and damaging de novo mutations as described above and in the Supplementary Note.

**Definition of known developmental disorder-associated genes**
To obtain a list of 'known' ARDD genes, we combined the list of genes from the DDG2P used by DDD with a list of diagnostic genes used within

the GeneDx in-house pipelines. There is an additional list of candidate genes in which GeneDx reports variants of unknown significance; we do not consider these here. We downloaded the latest version of DDG2P on 6 March 2023 from https://www.ebi.ac.uk/gene2phenotype/downloads and retained those genes listed as having 'definitive', 'strong' and 'moderate' evidence (that is, the clinically reportable categories), which were listed as 'biallelic_autosomal'. This left 1,236 genes. From the GeneDx list (current July 2022), we retained those annotated as 'validated' (that is, the clinically reportable categories), which were listed as 'autosomal recessive'. The GeneDx gene curation rules consider the following evidence in the course of validating a disease gene: replication (at least two independent publications or one large collaborative paper recruiting individuals from different backgrounds if GeneDx was involved) and the number of probands segregating molecularly strong variants. There were 2,223 autosomal recessive genes on the 'validated' list, many of which are actually associated with disorders that are not developmental disorders. Of these, we retained the 1,144 that were also 'biallelic_autosomal' on the DDG2P list (at any confidence level), plus 331 that were not on the DDG2P ARDD gene list but were classed as autism or intellectual disability genes by GeneDx. The remaining 748 genes were curated by two clinical geneticists (H.V.F. and E.S.) to determine those that caused developmental disorders as opposed to later-onset disorders and, of these, 191 were retained. The GeneDx ARDD gene list thus contained 1,666 genes. We called the 1,074 genes present on both the DDG2P and GeneDx ARDD gene lists 'consensus' genes, and the 754 present on just one of those lists 'discordant' genes; of these, 1,069 and 749, respectively were among the 17,320 genes retained for analysis (Supplementary Table 5).

**Phenotypic similarity of patients**
The phenotypic similarity of patients was calculated following a previous publication[6] with the *phenopy* package https://github.com/GeneDx/phenopy. Further details are given in the Supplementary Note.

**Reporting summary**
Further information on research design is available in the Nature Portfolio Reporting Summary linked to this article.

## Data availability
Sequence and variant-level data and phenotype data from the DDD study data are available on the European Genome-phenome Archive (EGA; https://www.ebi.ac.uk/ega) under study ID EGAS00001000775. The datasets required for replicating findings in this study are EGAD00001004388, EGAD00001004389 and EGAD00001004390. GeneDx data cannot be made available through the EGA owing to the nature of consent for clinical testing. GeneDx-referred patients are consented for aggregate, deidentified research and subject to US Health Insurance Portability and Accountability Act (HIPAA) privacy protection. As such, we are not able to share patient-level BAM or VCF data, which are potentially identifiable without a HIPAA Business Associate Agreement. GeneDx is a covered entity subject to HIPAA regulations regarding the use and disclosure of protected health information. GeneDx may not share patient-level BAM or VCF data, which may identify a patient, without first having a HIPAA business associate agreement or other legally required agreement in place between GeneDx and the requestor. Requestors must meet all applicable HIPAA requirements regarding the access, use, disclosure and storage of the data. Upon execution of all necessary documentation, patient-level data will be provided in accordance with the terms of the agreement between GeneDx and the requestor. Access to the deidentified aggregate data used in this analysis is available upon request to GeneDx (support@genedx.com). Requests will typically be fulfilled within 60 days. GeneDx has contributed deidentified data to this study to improve clinical interpretation of genomic data, in accordance with

patient consent and in conformance with the ACMG position statement on genomic data sharing. Clinically interpreted variants and associated phenotypes from the DDD study are available through DECIPHER (https://www.deciphergenomics.org). Clinically interpreted variants from GeneDx are deposited in ClinVar (https://www.ncbi.nlm.nih.gov/clinvar) under organization ID 26957 (https://www.ncbi.nlm.nih.gov/clinvar/submitters/26957). We used the GRCh37 reference genome, available at https://ftp.ensembl.org/pub/grch37/current/fasta/homo_sapiens/dna, 1000 Genomes phase 3 data, available at http://ftp.1000genomes.ebi.ac.uk/vol1/ftp/release/20130502, HGDP data, available at ftp://ngs.sanger.ac.uk/production/hgdp and gnomAD v.2 data, available at https://gnomad.broadinstitute.org/downloads.

## Code availability

The code to perform the burden analysis and reproduce plots from this paper is available on GitHub (https://github.com/chundruv/DDD_GeneDx_Recessives) or Zenodo (https://doi.org/10.5281/zenodo.12685780)[54], as is the code to run the phenopy method (https://github.com/GeneDx/phenopy).

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

## Acknowledgements

The DDD study presents independent research commissioned by the Health Innovation Challenge Fund (grant number HICF-1009-003). This study makes use of DECIPHER, which is funded by the Wellcome Trust. The full acknowledgements can be found at https://www.ddduk.org/accessing-ddd-data/. We additionally thank R. Hobson, E. Delage and the Human Genetics Informatics Team at Sanger for their input on the DDD study. This research was funded in whole or in part by the Wellcome Trust Grant 220540/Z/20/A, 'Wellcome Sanger Institute Quinquennial Review 2021–2026', Wellcome Trust Grant 226083/Z/22/Z and was supported by the National Institute for Health and Care Research (NIHR) Exeter Biomedical Research Centre. The views expressed are those of the author(s) and not necessarily those of the NIHR or the Department of Health and Social Care. For the purpose of Open Access, the author has applied a CC BY public copyright license to any Author Accepted Manuscript version arising from this submission. D.S.M. is supported by a Gates Cambridge Scholarship (OPP1144).

## Author contributions

V.K.C., Z.Z., K.W., S.L., R.T., H.O., V.D.U. and H.C.M. analyzed the data. P.D., R.Y.E., E.J.G., D.S.M., E.M.W. and K.S. contributed to code, methods and quality control analyses. R.T., K.R., C.F.W., M.E.H., K.E.S., V.D.U. and H.C.M. supervised the study. A.A., I.H.A., D.T., A.I.A.B., P.B., E.S.R., H.V.F., B.R., M.J.G.S., A.B.A., K.M. and E.S. provided clinical input. H.C.M. conceived and directed the study. V.K.C. and H.C.M. wrote the first draft of the manuscript, with input from Z.Z., K.W. and V.D.U. All authors commented on the final manuscript.

## Competing interests

K.M., M.J.G.S., H.O. and V.D.U. are employees of GeneDx. Z.Z., K.R. and R.T. were formerly employees of GeneDx and K.R. and R.T. are now employees of Geisinger Health System. A.B.A. and P.B. are employees of CENTOGENE. E.J.G. is an employee of and holds shares in Adrestia Therapeutics. K.E.S. has received support from Microsoft for work related to rare disease diagnostics. M.E.H. is a co-founder of, consultant to and holds shares in Congenica, a genetics diagnostic company. All other authors declare no conflicts of interest.

## Additional information

**Extended data** is available for this paper at https://doi.org/10.1038/s41588-024-01910-8.

**Correspondence and requests for materials** should be addressed to Hilary C. Martin.

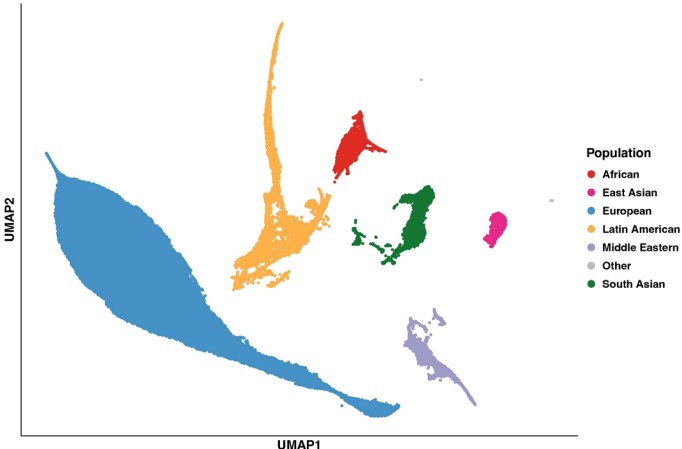

**Extended Data Fig. 1 | Defining broad-scale population structure.** UMAP of the first seven principal components (PCs) of the 1000 Genomes and HGDP samples with DDD and GeneDx samples projected onto the PCs. The genetically-inferred ancestry (GIA) groups were labelled based on the ancestry of the 1000 Genomes/HGDP reference samples within each cluster.

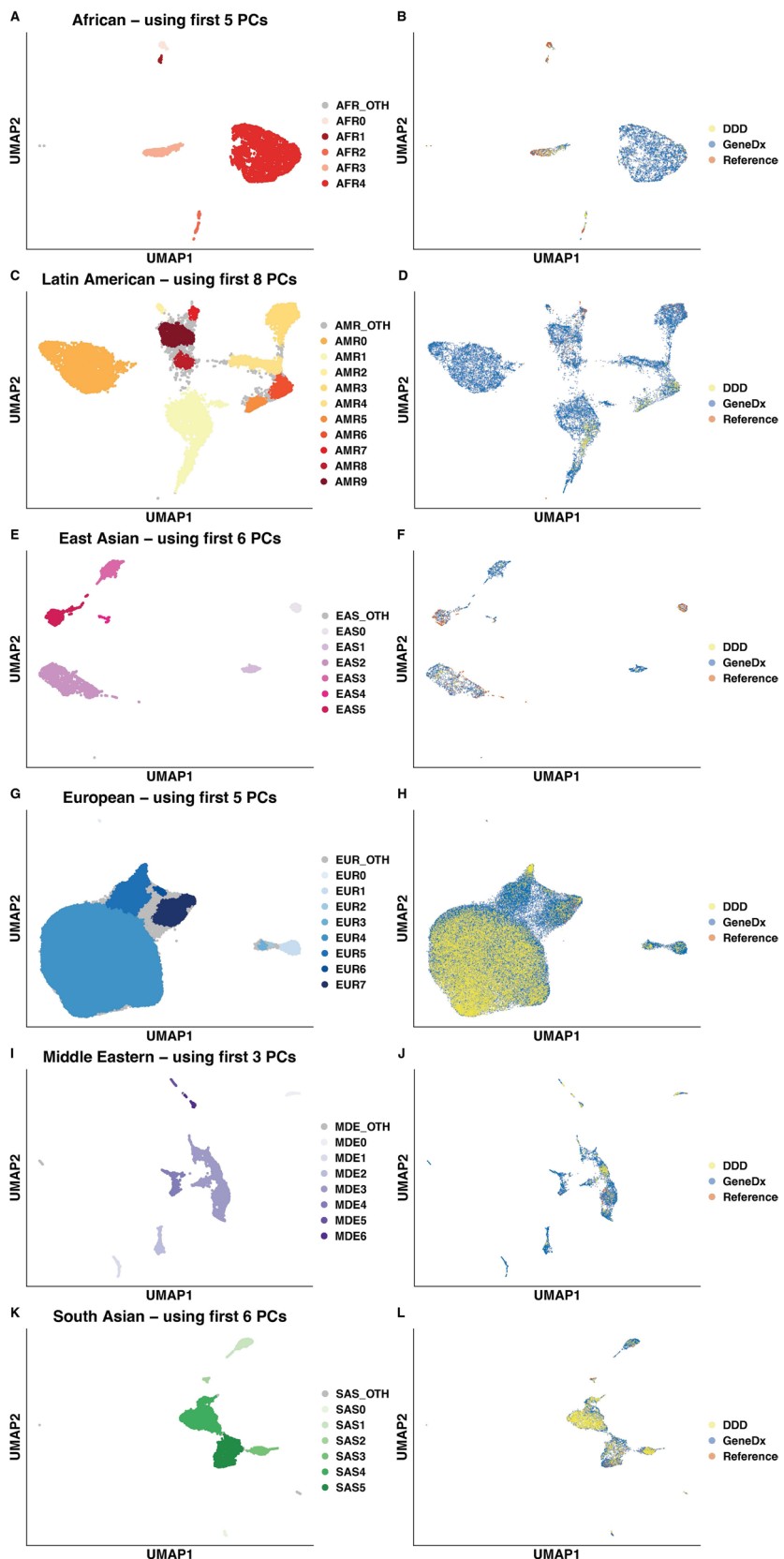

**Extended Data Fig. 2 | Defining fine-scale population structure.** UMAPs based on principal components from each continental-level genetically-inferred ancestry (GIA) group from Extended Data Fig. 1. The PCA was run on each GIA group separately using the 1000 Genomes/HGDP reference samples together with the unrelated parents from GeneDx, then the DDD samples and remaining GeneDx samples were projected onto these. The clusters indicated in the left-hand plots were determined using HDBSCAN. The right-hand plots show the same UMAP but instead coloured to indicate which samples come from each cohort versus the reference samples. The GIA groups were as follows: **A–B)** African (AFR), **C–D)** Latin American (AMR), **E–F)** East Asian (EAS), **G–H)** European (EUR), **I–J)** Middle Eastern (MDE), and **K–L)** South Asian (SAS).

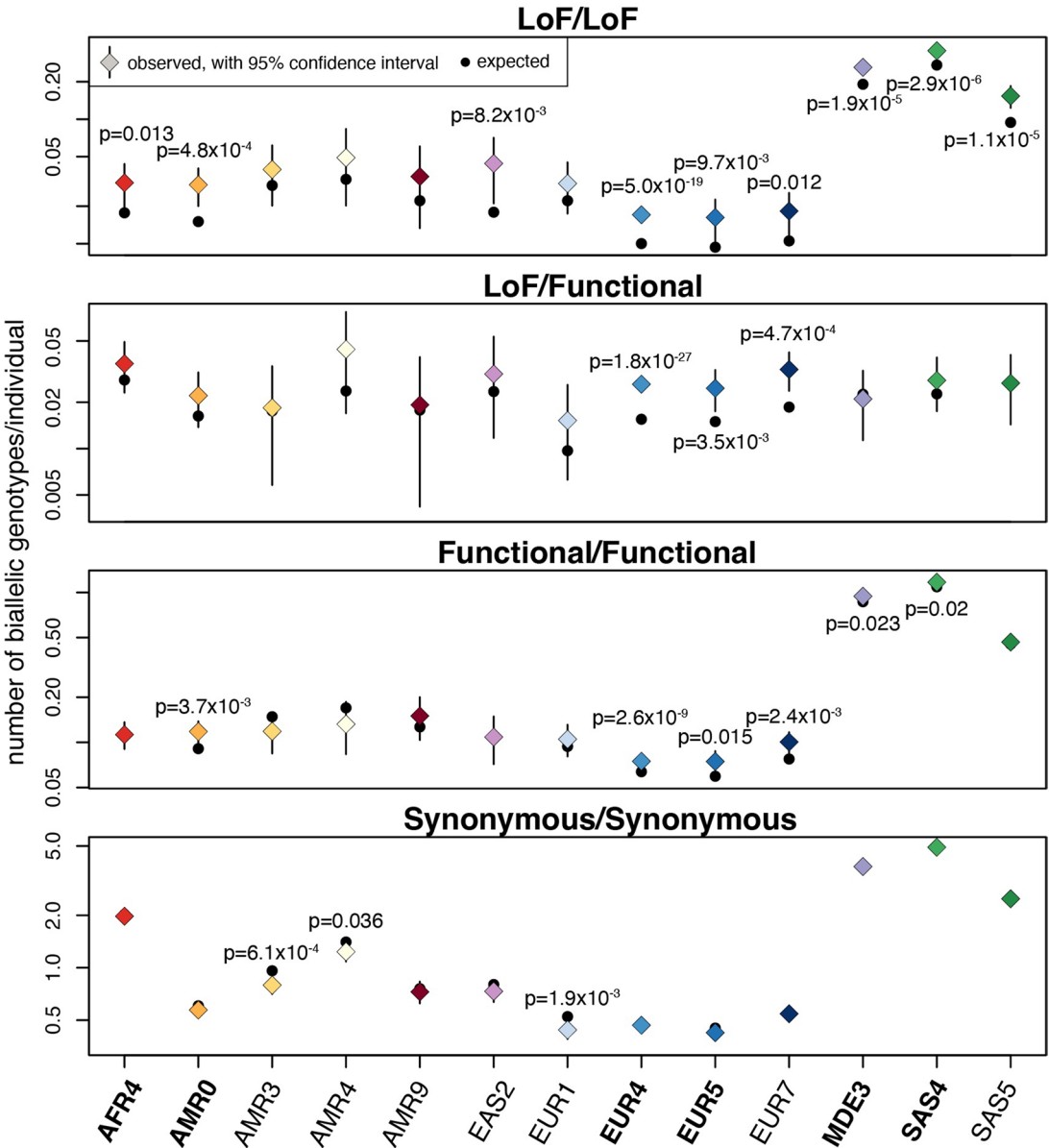

**Extended Data Fig. 3 | Exome-wide observed and expected number of biallelic genotypes per genetically-inferred ancestry (GIA) sub-group, for the four consequence classes.** This is after excluding trios with cross-continental admixture. This figure shows only GIA sub-groups with at least 200 trios; numbers for all GIA sub-groups are shown in Supplementary Table 4, together with estimates obtained with either no admixture filtering or stricter admixture filtering. The GIA sub-groups used in Fig. 1 are shown in blue bold text along the x-axis. Coloured points are the observed numbers, black points are the expected numbers, and black lines show 95% confidence intervals around the observed. For some GIA sub-groups, the black points and/or black lines are not visible as they lie under the coloured points. P-values are shown for those where there is a Bonferroni significant difference between the observed and expected values, according to a Poisson test ($p < 0.05/88$, since in total there were 4 tests from each of 22 populations; two-sided test for synonymous/synonymous, one-sided otherwise).

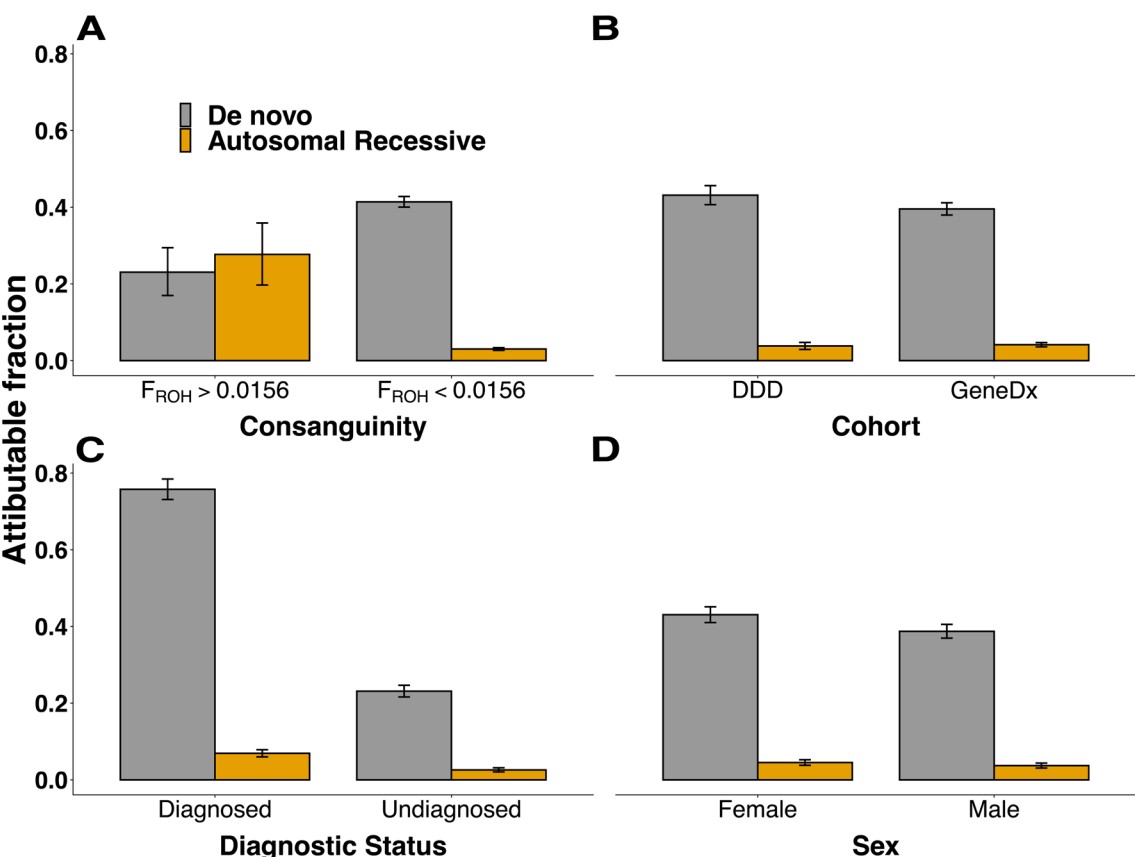

**Extended Data Fig. 4 | *De novo* or autosomal recessive attributable fraction in different subsets of probands.** Fraction of patients in different groups attributable to *de novo* versus autosomal recessive coding variants [(observed-expected)/N]. The patients are split by (**a**) level of consanguinity (N = 1,087 and 24,436 for low and high consanguinity respectively), (**b**) cohort (N = 7,919 and 17,604 for DDD and GeneDx respectively), (**c**) diagnostic status (N = 8,494 and 17,029 for diagnosed and undiagnosed respectively) or (**d**) sex (N = 11,316 and 14,207 for female and male respectively). The bars show the attributable fraction estimates within the groups, and error bars show 95% confidence intervals.

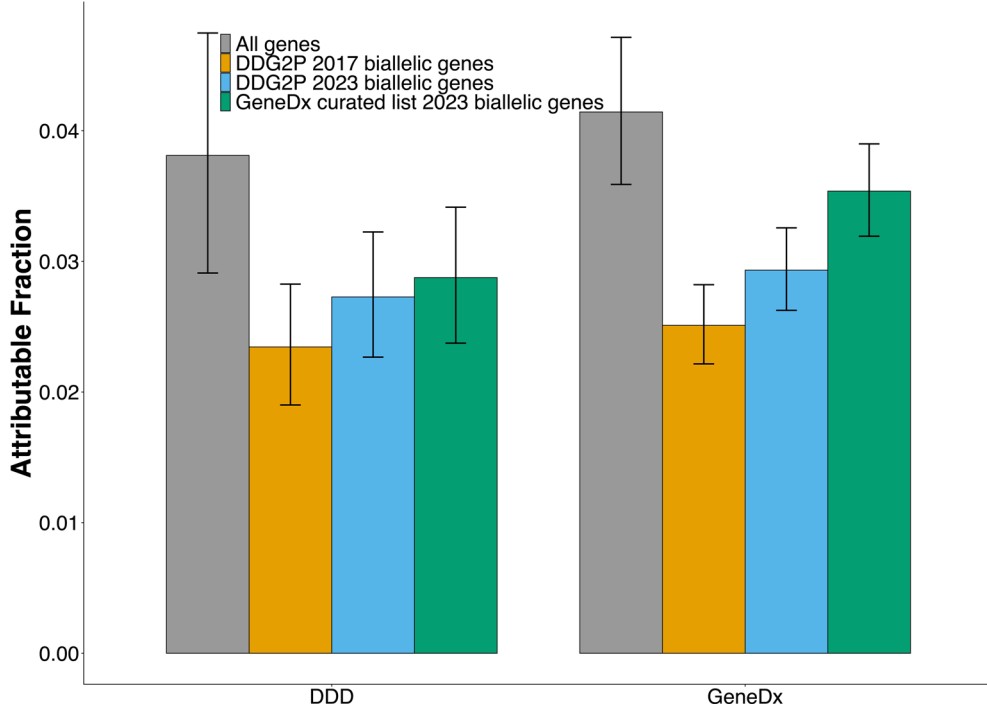

**Extended Data Fig. 5 | Autosomal recessive attributable fraction in different gene lists.** Fraction of patients in each cohort attributable to autosomal recessive coding variants both across all genes and in the indicated ARDD gene lists (N = 7,919 and 17,604 for DDD and GeneDx respectively). Error bars show 95% confidence intervals.

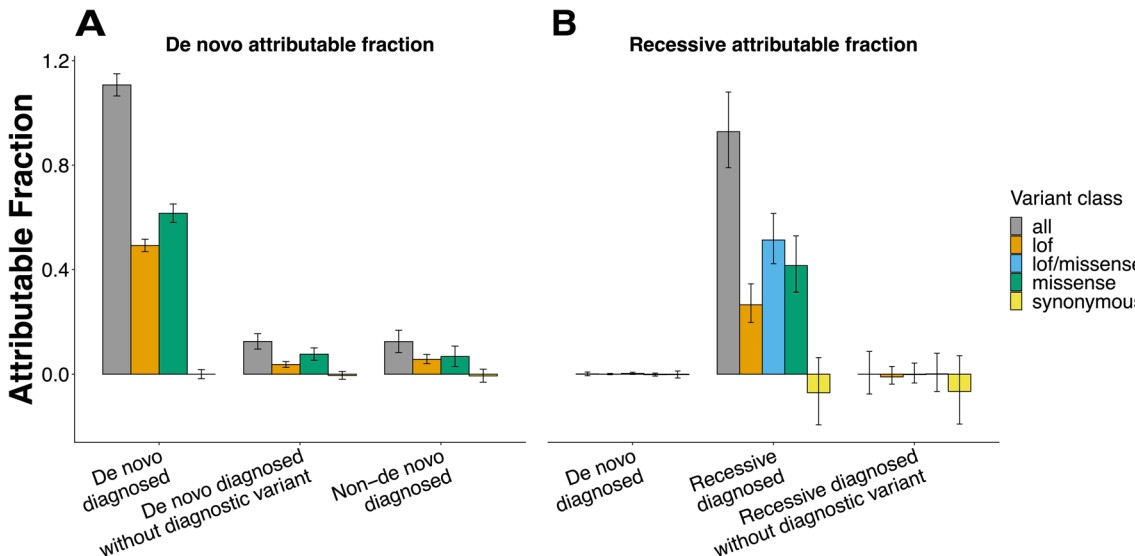

**Extended Data Fig. 6 | Quantifying the contribution of as-yet-undetected multi-gene diagnoses.** A) The residual *de novo* and recessive attributable fraction in diagnosed individuals before and after diagnostic variants were removed (N = 2,031 and 4,624 diagnosed with non-*de novo* and *de novo* respectively). B) The residual *de novo* attributable fraction in diagnosed patients, excluding the diagnostic variant, restricted to monoallelic or X-linked dominant DDG2P genes versus all other genes. Note the patients with partial diagnoses in DDD were included but patients with known composite diagnoses or whose diagnostic variant/s did not pass variant filters were excluded from the diagnosed sets. Error bars show 95% confidence intervals.

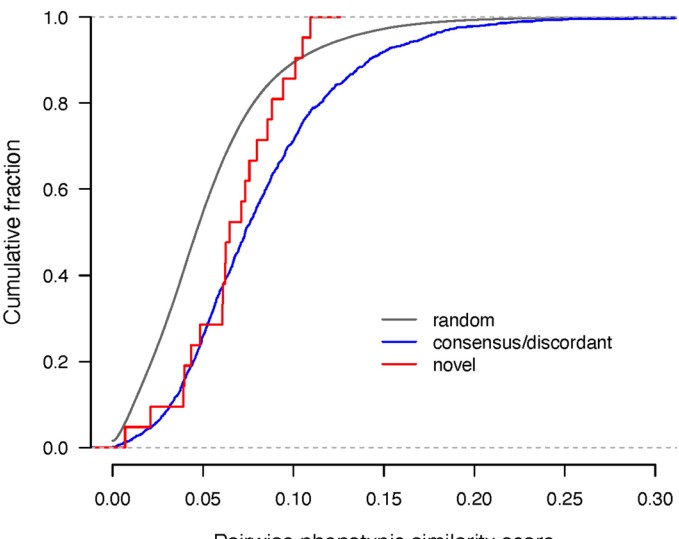

**DDD and GeneDx combined**

**Extended Data Fig. 7 | Assessing phenotypic similarity between patients with biallelic genotypes in the same gene.** Cumulative distribution functions for pairwise phenotypic similarity scores as calculated by Phenopy. The distribution of novel genes passing FDR < 5% (*ATAD2B*, *CRELD1*, *HECTD4*, *KBTBD2*, *ZDHHC16*) is shown in red, consensus/discordant genes passing FDR < 5% in blue, and the similarity scores of random pairs in grey. Random pairs were selected proportionally to match the occurrence of DDD/DDD, GeneDx/GeneDx and DDD/GeneDx pairs in the novel and consensus/discordant sets. The phenotypic similarity scores in patients with damaging biallelic genotypes in the novel genes were not significantly lower than those for patients with such genotypes in consensus/discordant genes (one-sided Wilcoxon rank sum p = 0.12), but they were significantly higher than random scores (one-sided Wilcoxon rank sum p = 0.0058).

|---|---|

# Reporting Summary

## Statistics

For all statistical analyses, confirm that the following items are present in the figure legend, table legend, main text, or Methods section.

| n/a | Confirmed | |
|---|---|---|
| ☐ | ☒ | The exact sample size (*n*) for each experimental group/condition, given as a discrete number and unit of measurement |
| ☒ | ☐ | A statement on whether measurements were taken from distinct samples or whether the same sample was measured repeatedly |
| ☐ | ☒ | The statistical test(s) used AND whether they are one- or two-sided<br>*Only common tests should be described solely by name; describe more complex techniques in the Methods section.* |
| ☐ | ☒ | A description of all covariates tested |
| ☐ | ☒ | A description of any assumptions or corrections, such as tests of normality and adjustment for multiple comparisons |
| ☐ | ☒ | A full description of the statistical parameters including central tendency (e.g. means) or other basic estimates (e.g. regression coefficient) AND variation (e.g. standard deviation) or associated estimates of uncertainty (e.g. confidence intervals) |
| ☐ | ☒ | For null hypothesis testing, the test statistic (e.g. *F*, *t*, *r*) with confidence intervals, effect sizes, degrees of freedom and *P* value noted<br>*Give P values as exact values whenever suitable.* |
| ☒ | ☐ | For Bayesian analysis, information on the choice of priors and Markov chain Monte Carlo settings |
| ☒ | ☐ | For hierarchical and complex designs, identification of the appropriate level for tests and full reporting of outcomes |
| ☐ | ☒ | Estimates of effect sizes (e.g. Cohen's *d*, Pearson's *r*), indicating how they were calculated |

*Our web collection on statistics for biologists contains articles on many of the points above.*

## Software and code

Policy information about availability of computer code

| Data collection | No software was used to collect data. |
|---|---|
| Data analysis | In DDD, the BWA aln algorithm (BWA version 0.5.10) and the BWA mem algorithm (BWA version 0.7.12) were used to align reads.Picard Markduplicates (versions 1.98 and 1.114)33 and Genome Analysis Toolkit IndelRealigner (GATK version 3.1.1 and version 3.5.0) were used for sample-level BAM improvement. GATK (version 3.5.0) HaplotypeCaller was used for variant calling.<br><br>In GeneDx, BWA (v0.5.8 to v0.7.8, depending on the time of sequencing) was used to align the reads. BAM files were then converted to CRAM format with Samtools version 1.3.139. Individual gVCF files were called with GATK v3.7-0 HaplotypeCaller. Multi-sample GVCF files were jointly genotyped using GATK v3.7-0 GenotypeGVCFs. GATK v3.7-0 VariantRecalibrator (VQSR) was applied for both SNPs and indels.<br><br>KING (version 2.2) was used to infer kinship. GCTA (version 1.93.0) was used to run principal component analysis, and HBDSCAN to determine fine- scale ancestry clusters. Variant annotation was carried out using VEP v94.5.<br><br>We made use of the following software and packages:<br>R 3.6.3(for analysis) & 4.1.3(for plotting)<br>Python 3.9.15<br>htslib 1.15.1<br>bcftools 1.16-63<br>r-dplyr 1.0.6<br>tabix 1.11<br>numpy 1.24.1 |

| scipy | 1.10.0 |
| cyvcf2 | 0.30.16 |
| pandas | 1.5.2 |
| pytabix | 0.1 |
| pyranges | 0.0.120 |
| umap-learn | 0.5.3 |
| hdbscan | 0.8.29 |
| r-ggplot2 | 3.4.2 |
| r-dplyr | 1.1.0 |
| r-cowplot | 1.1.1 |
| r-data.table | 1.14.6 |
| r-ggridges | 0.5.4 |
| r-patchwork | 1.1.2 |
| plink | 1.9 |

The code to perform the burden analysis and reproduce plots from this paper is available on GitHub (https://github.com/chundruv/ DDD_GeneDx_Recessives), as is the code to run the Phenopy method (https://github.com/GeneDx/phenopy).

We made use of data from gnomAD v2.1.1, 1000 Genomes Phase 3 and Human Genome Diversity Project (2019 release).

For manuscripts utilizing custom algorithms or software that are central to the research but not yet described in published literature, software must be made available to editors and reviewers. We strongly encourage code deposition in a community repository (e.g. GitHub). See the Nature Portfolio guidelines for submitting code & software for further information.

## Data

Policy information about availability of data

All manuscripts must include a data availability statement. This statement should provide the following information, where applicable:
- Accession codes, unique identifiers, or web links for publicly available datasets
- A description of any restrictions on data availability
- For clinical datasets or third party data, please ensure that the statement adheres to our policy

Sequence and variant-level data and phenotype data from the DDD study data are available on the European Genome-phenome Archive (EGA; https://www.ebi.ac.uk/ega/) with study ID EGAS00001000775. The datasets most of interest for replicating findings in this study are EGAD00001004388, EGAD00001004389 and EGAD00001004390. GeneDx data cannot be made available through the EGA owing to the nature of consent for clinical testing. GeneDx-referred patients are consented for aggregate, deidentified research and subject to US HIPAA privacy protection. As such, we are not able to share patient-level BAM or VCF data, which are potentially identifiable without a HIPAA Business Associate Agreement. Access to the deidentified aggregate data used in this analysis is available upon request to GeneDx. GeneDx has contributed deidentified data to this study to improve clinical interpretation of genomic data, in accordance with patient consent and in conformance with the ACMG position statement on genomic data sharing. Clinically interpreted variants and associated phenotypes from the DDD study are available through DECIPHER (https://www.deciphergenomics.org/). Clinically interpreted variants from GeneDx are deposited in ClinVar (https://www.ncbi.nlm.nih.gov/clinvar) under organisation ID 26957 (https://www.ncbi.nlm.nih.gov/clinvar/submitters/26957/).

## Research involving human participants, their data, or biological material

Policy information about studies with human participants or human data. See also policy information about sex, gender (identity/presentation), and sexual orientation and race, ethnicity and racism.

| Reporting on sex and gender | Only one analysis in this paper (Supplementary Figure 11) is stratified by sex. It was stratified by genetically-determined sex. The purpose was to test for sex differences in the recessive burden of developmental disorders |
| Reporting on race, ethnicity, or other socially relevant groupings | In this work, we classified individuals into genetically-inferred ancestry (GIA) groups. The rationale for this was two-fold: firstly, we were interested in exploring differences in genetic architecture between these groups, and secondly, the analysis relies on accurate estimates of allele frequencies that differ between groups. The classifications were based on genetic similarity to individuals in the 1000 Genomes and Human Genome Diversity Panel (HGDP) reference datasets, inferred from principal component analysis. We defined six continental-level GIA groups (AFR: African; AMR: Latin American; EAS: East Asian; EUR: European; MDE: Middle Eastern; SAS: South Asian) and, within these, forty-seven fine-scale GIA sub-groups. The Methods are fully described in the manuscript, as is the justification for doing this. |
| Population characteristics | The individuals in this study are patients with neurodevelopmental disorders (or their parents) who were genetically undiagnosed prior to sequencing by the DDD study or GeneDx. For DDD, the average age was 7.3 (standard deviation 6.1 years). For GeneDx, it was 9.4 years (SD 10.2 years). 58.4% of DDD probands were male versus 55.7% in GeneDx. |
| Recruitment | Between April 2011 and April 2015, the DDD study patients a severe developmental disorder who remained undiagnosed after undergoing the typical clinical genetics investigations. The phenotypic inclusion criteria included neurodevelopmental disorders, congenital abnormalities, growth abnormalities, dysmorphic features, and unusual behavioural phenotypes. Recruitment took place across twenty-four regional genetics services within the United Kingdom and the Republic of Ireland health services. Since the study only included patients who were undiagnosed through the usual clinical means, it may be depleted of easy-to-diagnose recessive cases, as mentioned in the main text. Patients were referred to GeneDx for clinical whole-exome sequencing for diagnosis of suspected Mendelian disorders, as described in https://www.nature.com/articles/gim2015148. Patients were selected for inclusion in this study based on having one or more HPO terms from a list of 716 that fell under "abnormality of the nervous system". |
| Ethics oversight | The DDD study was approved by the UK Research Ethics Committee (10/H0305/83, granted by the Cambridge South Research Ethics Committee, and GEN/284/12, granted by the Republic of Ireland Research Ethics Committee). The GeneDx |

study was conducted in accordance with all guidelines set forth by the Western Institutional Review Board, Puyallup, WA (WIRB 20162523).

Note that full information on the approval of the study protocol must also be provided in the manuscript.

# Field-specific reporting

Please select the one below that is the best fit for your research. If you are not sure, read the appropriate sections before making your selection.

☒ Life sciences          ☐ Behavioural & social sciences          ☐ Ecological, evolutionary & environmental sciences

For a reference copy of the document with all sections, see nature.com/documents/nr-reporting-summary-flat.pdf

# Life sciences study design

All studies must disclose on these points even when the disclosure is negative.

| | |
|---|---|
| Sample size | We used the largest available sample size, combining data from two cohorts. |
| Data exclusions | We excluded individuals who did not have neurodevelopmental disorders, or who failed genetic quality control, as described in the Methods section. |
| Replication | We did not replicate as each disorder is very rare, as such we did not have another dataset to use. We did however find extra cases for the significant genes |
| Randomization | No randomization was needed. We also did not control for covariates (e.g. genetic principal components), but rather restricted the calculation of allele frequencies to genetically homogeneous subgroups. |
| Blinding | Not applicable since no specific grouping. |

# Reporting for specific materials, systems and methods

We require information from authors about some types of materials, experimental systems and methods used in many studies. Here, indicate whether each material, system or method listed is relevant to your study. If you are not sure if a list item applies to your research, read the appropriate section before selecting a response.

## Materials & experimental systems

| n/a | Involved in the study |
|---|---|
| ☒ | ☐ Antibodies |
| ☒ | ☐ Eukaryotic cell lines |
| ☒ | ☐ Palaeontology and archaeology |
| ☒ | ☐ Animals and other organisms |
| ☒ | ☐ Clinical data |
| ☒ | ☐ Dual use research of concern |
| ☒ | ☐ Plants |

## Methods

| n/a | Involved in the study |
|---|---|
| ☒ | ☐ ChIP-seq |
| ☒ | ☐ Flow cytometry |
| ☒ | ☐ MRI-based neuroimaging |

