## [Peer Review File · Nature Genetics]

Peer Review Information

Manuscript Title: Federated analysis of autosomal recessive coding variants in 29,745 developmental disorder patients from diverse populations

Corresponding author name(s): Dr Hilary (C) Martin

Reviewer Comments & Decisions:

Decision Letter, initial version:

16th November 2023

Dear Hilary,

Your Article "Federated analysis of the contribution of recessive coding variants to 29,745 developmental disorder patients from diverse populations" has been seen by two referees. You will see from their comments below that, while they find your work of potential interest, they have raised concerns that must be addressed. In light of these comments, we cannot accept the manuscript for publication at this time, but we would be interested in considering a suitably revised version that addresses the referees' concerns.

We hope you will find the referees' comments useful as you decide how to proceed. If you wish to submit a substantially revised manuscript, please bear in mind that we will be reluctant to approach the referees again in the absence of major revisions.

To guide the scope of the revisions, the editors discuss the referee reports in detail within the team, including with the chief editor, with a view to identifying key priorities that should be addressed in revision, and sometimes overruling referee requests that are deemed beyond the scope of the current study. In this case, we ask that you thoroughly address all concerns related to variant filtering and the evidence supporting the newly implicated candidate genes, providing additional supporting evidence where feasible and/or revising interpretations where appropriate, and that you extend the analyses to consider X-linked genes and oligogenic inheritance as requested by Reviewer #1. We hope you will find this prioritized set of referee points to be useful when revising your study. Please do not hesitate to get in touch if you would like to discuss these issues further.

If you choose to revise your manuscript taking into account all reviewer and editor comments, please highlight all changes in the manuscript text file. At this stage, we will need you to upload a copy of the manuscript in MS Word .docx or similar editable format.

*2) If you have not done so already, please begin to revise your manuscript so that it conforms to our Article format instructions, available here. Refer also to any guidelines provided in this letter.

*3) Include a revised version of any required Reporting Summary: <https://www.nature.com/documents/nr-reporting-summary.pdf>
It will be available to referees (and potentially statisticians) to aid in their evaluation if the manuscript goes back for peer review.
A revised checklist is essential for re-review of the paper.

Please be aware of our guidelines on digital image standards.

[redacted]

If you wish to submit a suitably revised manuscript, we hope to receive it within 3-6 months. If you cannot send it within this time, please let us know. We will be happy to consider your revision so long as nothing similar has been accepted for publication at Nature Genetics or published elsewhere. Should your manuscript be substantially delayed without notifying us in advance and your article is eventually published, the received date would be that of the revised, not the original, version.

Nature Genetics is committed to improving transparency in authorship. As part of our efforts in this direction, we are now requesting that all authors identified as 'corresponding author' on published papers create and link their Open Researcher and Contributor Identifier (ORCID) with their account on the Manuscript Tracking System (MTS), prior to acceptance. ORCID helps the scientific community achieve unambiguous attribution of all scholarly contributions. You can create and link your ORCID from the home page of the MTS by clicking on 'Modify my Springer Nature account'. For more information, please visit www.springernature.com/orcid.

Thank you for the opportunity to review your work.

Sincerely,
Kyle

Kyle Vogan, PhD
Senior Editor
Nature Genetics
<https://orcid.org/0000-0001-9565-9665>

Referee expertise:

Referee #1: Genetics, developmental disorders

Referee #2: Genetics, neurodevelopmental disorders

Reviewers' Comments:

Reviewer #1:
Remarks to the Author:

This manuscript represents a largest-of-its-kind study to date, which leveraged nearly 30k parent-proband trios with developmental disorders (DD) to characterize the contribution of autosomal recessive (AR) coding variants to disease. This was a massive undertaking and involved merging two of the largest known such datasets, the DDD and GeneDx trio data, to tackle this question. Important findings from the manuscript include the following: (1) the fraction of probands with AR coding variants was significantly correlated with the average autozygosity; (2) established AR DD-associated genes explained 90% of the total AR coding burden, which was a substantial increase from a previous study from these authors 5 years ago (Martin et al. 2018 Science; estimated at ~48%); (3) there was no significant difference between probands of European vs non-European ancestry, which is reassuring given the persistent skewing of control databases for European exomes and genomes; (4) the authors estimate that ~1% of undiagnosed probands are underpinned by missense variants, thus highlighting a lingering interpretive challenge. Finally, (5) the authors performed gene-specific enrichment of damaging biallelic genotypes, identifying multiple genes (n=25) that pass Bonferroni correction; 9 genes including KBTBD2 (2 affected individuals) and CRELD1 (8 affected individuals) are not on any known ARDD gene list and are nominated as novel causal genes.

Major strengths of this manuscript involved the numbers of individuals included at 29,745 trios; careful consideration of phenotypes fitting a "nervous system" HPO term; classification of individuals into genetically inferred ancestry groups; quality control of exome data across cohorts; and thoughtful use of lists of known ARDD disease genes. Importantly, a substantial fraction of affected individuals were of non-European ancestry (~20%), and enabled comparison of the contribution of AR coding variants across ancestry groups, which brings necessary attention to understudied ethnic groups. Although some of the results are not particularly surprising (e.g. correlation between autozygosity and AR coding variant contribution), it is useful to have the main messages of the paper supported with the large patient numbers and ethnic diversity. The manuscript also takes a pulse on the ARDD

diagnostic rate and indicates that the field has made marked progress in the past several years but is reaching saturation by using a gene-specific enrichment approach to identify new causal disease genes.

I also appreciate (and mostly agree with) the candor of the authors as they list the limitations of the study in the Discussion (non-random sampling; assumed simple genetic architecture; non-representation of the global population; lack of inclusion of CNVs or non-coding variants). I do not feel strongly that efforts to overcome the above limitations are within the capability of the existing datasets or the scope of the current work. My only major concerns with the manuscript hinge on the following opportunities missed:

1. X-linked (XL) contribution to DD. The authors have existing trio data, information about male/female, and known causal gene lists. Although they show nicely in Figure 1 the contribution of de novo and AR coding variants, the manuscript is lacking information about the coding variant contribution of XL genes, many of which contribute to DD. It would be crucial to determine whether there exist differences (or not) for Europeans vs non-Europeans.

2. Multi-gene vs monogenic causes of DD. Multiple lines of evidence from the literature point toward affected individuals with DD caused by two different genes (and not just a single-gene cause). Can this large dataset be leveraged to understand not only the fraction of variants attributable to AR coding alleles, but the incidence of individuals with a single vs multiple genes contributing to disease? There is potentially a hint of this phenomenon in Table 2 with the "contrary evidence" of individuals who have confirmed diagnoses in alternative genes (ATG4C, LRRRC34, and C11ORF94).

3. Gene discovery. Table 2 lists the nine genes not on any known AR DD list, but that pass Bonferroni correction.

- a. Can the "level of evidence" information be defined by more objective criteria than "very strong", "strongly suggestive" or "weakly suggestive"?

- b. Further information discussing CRELD1 is warranted, particularly explanation of how dominant variants give rise to a markedly different phenotype than AR variants.

- c. Nearly half of genes in Table 2 are "weakly suggestive" (n=4), and this is disappointing given that these are highlighted among the most likely new causal ARDD genes to come out of these analyses. Further evidence in support of these genes as new ARDD loci is needed (all of them with 2 affected individuals or less), whether it be in the form of additional cases (identified outside of the DDD or GeneDx datasets through gene matching); or through the use of vertebrate animal models to show relevance to DD phenotypes.

- d. Phenotype similarity is difficult to appreciate for individuals listed in Supplementary Table 7. It would be extremely helpful to have individual HPO terms listed by column and indicators of present/absent/no data for each individual so that overlap can be better visualized.

Reviewer #2:
Remarks to the Author:

With this study, the authors expand on a previous one, published in 2018, where they estimated the genome-wide contribution of recessive variants in 6040 families from the Deciphering Developmental Disorders study. The same objective led them to combine this data with new one from patients of the genetic diagnostics company GeneDx who had some “abnormality of the nervous system,” thus creating a new cohort of 29,745 patients. They created genetically inferred ancestry (GIA) groups and sub-groups with little significant outcome. They identified 2 novel autosomal recessive developmental disorders (ARDD) genes, KBTBD2 and CRELD1, and found suggestive evidence for ZDHHC16, ATG4C, HECTD4, ATAD2B, ATXN1, LRRC34, and C11ORF94. They also highlighted the challenges of such approach.

I have major and minor comments.

Major comments:

It would be useful if the authors discussed the function of the proteins, and ideally the expression pattern, encoded by the genes they suggest are new recessive disease genes. For example, C11ORF94 seems important for spermatids and seems expressed specifically in the testes, this raises concerns for its implication in DD. More on this gene in the following comment.

I am concerned by the stringency of the filtering for the variants. For example, ATG4C is tolerant to LoF variants. Variant chr1-63299729-TTG-T found in 5 of 7 patients, some at homozygosity, is carried by 1/70 European, and is found at homozygosity in multiple individuals in the gnomAD v2.1.1 non-neuro cohort. Variants in CRELD1, LRRC34 and C11ORF94 especially also have non-negligible MAFs in gnomAD and such variants are present in 50% or more of the patients for these 3 genes. For example, C11ORF94 is tolerant to LoF variants also, and one of the two variants (found at homozygosity in 1 of 2 patients), chr11-45928455-AG-AGG, when searched with a different nomenclature (chr11-45928455-A-AG), is found in about 1/300 Latinos.

Minor comments:

“We estimated that ~1% of undiagnosed patients in both cohorts (...)”: As the reader assumes that the authors are referring to the probands with genetically inferred European vs. non-European ancestries, this population segregation is not previously explained nor justified, especially considering that poorly significant results were obtained from GIA comparisons.

“We defined six continental-level GIA groups (...) and, within these, forty-seven fine-scale GIA sub-groups”: It is important to give some detail regarding the GIA sub-groups (e.g. based on a certain surface area, or population density, or ethnic distribution, or allele frequencies or autozygosity levels) elsewhere than in the supplementary data.

“This study (...) suggests that improving strategies for interpreting missense variants in known ARDD genes may allow us to diagnose more patients than discovering the remaining genes”: This conclusion seems quite evident. I would talk in addition about the limitations of a cohort of patients with complicated diagnosis, or relatively unselected cohorts, or other elements specific to this study’s cohort.

“clinically recognizable disorder (e.g.11,12)”: I would either say “clinically recognizable disorder

11,12” or “clinically recognizable disorder (e.g. Miller¹¹ and Wiedemann-Steiner¹² syndromes)”

“Forty-eight percent of the exome-wide burden of recessive causes was in known AR DD-associated (ARDD) genes, indicating that larger sample sizes would be required to find the additional genes”: As your study shows, larger cohorts are not necessarily more useful than better-curated cohorts.

“while the attributable fraction due to de novos was not”: “de novo variants” or “DNMs” instead

Author Rebuttal to Initial comments

We thank both reviewers for their helpful comments. Please note that important/substantial additions to the manuscript are indicated in blue text in the manuscript file.

Reviewers' Comments:

Reviewer #1:

Remarks to the Author:

This manuscript represents a largest-of-its-kind study to date, which leveraged nearly 30k parent-proband trios with developmental disorders (DD) to characterize the contribution of autosomal recessive (AR) coding variants to disease. This was a massive undertaking and involved merging two of the largest known such datasets, the DDD and GeneDx trio data, to tackle this question. Important findings from the manuscript include the following: (1) the fraction of probands with AR coding variants was significantly correlated with the average autozygosity; (2) established AR DD-associated genes explained 90% of the total AR coding burden, which was a substantial increase from a previous study from these authors 5 years ago (Martin et al. 2018 Science; estimated at ~48%); (3) there was no significant difference between probands of European vs non-European ancestry, which is reassuring given the persistent skewing of control databases for European exomes and genomes; (4) the authors estimate that ~1% of undiagnosed probands are underpinned by missense variants, thus highlighting a lingering interpretive challenge. Finally, (5) the authors performed gene-specific enrichment of damaging biallelic genotypes, identifying multiple genes (n=25) that pass Bonferroni correction; 9 genes including KBTBD2 (2 affected individuals) and CRELD1 (8 affected individuals) are not on any known ARDD gene list and are nominated as novel causal genes.

Major strengths of this manuscript involved the numbers of individuals included at 29,745 trios; careful consideration of phenotypes fitting a “nervous system” HPO term; classification of individuals into

genetically inferred ancestry groups; quality control of exome data across cohorts; and thoughtful use of lists of known ARDD disease genes. Importantly, a substantial fraction of affected individuals were of non-European ancestry (~20%), and enabled comparison of the contribution of AR coding variants across ancestry groups, which brings necessary attention to understudied ethnic groups. Although some of the results are not particularly surprising (e.g. correlation between autozygosity and AR coding variant contribution), it is useful to have the main messages of the paper supported with the large patient numbers and ethnic diversity. The manuscript also takes a pulse on the ARDD diagnostic rate and indicates that the field has made marked progress in the past several years but is reaching saturation by using a gene-specific enrichment approach to identify new causal disease genes.

We thank the reviewer for their positive comments and their careful reading of the manuscript.

I also appreciate (and mostly agree with) the candor of the authors as they list the limitations of the study in the Discussion (non-random sampling; assumed simple genetic architecture; non-representation of the global population; lack of inclusion of CNVs or non-coding variants). I do not feel strongly that efforts to overcome the above limitations are within the capability of the existing datasets or the scope of the current work. My only major concerns with the manuscript hinge on the following opportunities missed:

1. X-linked (XL) contribution to DD. The authors have existing trio data, information about male/female, and known causal gene lists. Although they show nicely in Figure 1 the contribution of de novo and AR coding variants, the manuscript is lacking information about the coding variant contribution of XL genes, many of which contribute to DD. It would be crucial to determine whether there exist differences (or not) for Europeans vs non-Europeans.

We thank the reviewer for their thoughtful comment. We have previously published on the contribution of X-linked coding variants to developmental disorders using the DDD data (Martin et al., *Nature Communications*, 2021). That paper showed that ~6% of both male and female probands could be explained by X-linked coding causes, and that the known X-linked developmental disorder genes explained ~80% of the chromosome-wide burden.

A major selling point about the current GeneDx+DDD dataset is its ancestral diversity. *A priori* we do not expect the rates of X-linked disorders to differ between ancestry groups, unlike the autosomal recessive contribution, which is strongly affected by autozygosity levels. Given this and the fact that our 2021 paper

showed that almost all the X-linked burden in DDD was in known developmental disorder genes, we do not believe that this analysis is likely to yield substantial new insights about the X-linked contribution to DD beyond those made in our previous paper. We believe that this extension is beyond the scope of this current paper. To reinforce this, we have slightly modified the paper's title to indicate that the focus is on "autosomal recessive" causes.

In addition to this scientific reasoning, there is a major practical reason why we are unable to fulfill the reviewer's request. Carrying out this analysis would require going back to square-one and carrying out chrX-specific QC of the GeneDx data, which would be a substantial amount of work. The key analyst at GeneDx (Zhancheng Zhang, second author of this manuscript) left the company more than six months ago, and the research team at GeneDx do not have the bandwidth to carry out this work. Given the smaller size of the DDD dataset, and the predominance of European ancestry in that cohort (~90%), we do not believe that DDD alone would have sufficient power to make meaningful comparisons of the contribution of chrX in Europeans and non-Europeans. Even if there is any difference, it is likely to be very small, resulting from the fact that the autosomal recessive burden is different (since the non-European group in DDD is dominated by South Asians with higher autozygosity) so the relative contribution of other mechanisms may also be slightly different; thus, we would probably not have the power to detect a significant difference. However, if the reviewer feels that a comparison of the chrX burden in Europeans versus non-Europeans in DDD alone would meaningfully improve this paper, we would be happy to carry this out.

2. Multi-gene vs monogenic causes of DD. Multiple lines of evidence from the literature point toward affected individuals with DD caused by two different genes (and not just a single-gene cause). Can this large dataset be leveraged to understand not only the fraction of variants attributable to AR coding alleles, but the incidence of individuals with a single vs multiple genes contributing to disease? There is potentially a hint of this phenomenon in Table 2 with the "contrary evidence" of individuals who have confirmed diagnoses in alternative genes (*ATG4C*, *LRR34*, and *C11ORF94*).

We thank the reviewer for this interesting suggestion, which we have been able to pursue using the data we already had in hand from GeneDx. Before responding more broadly, we would just like to note that we have now dropped *ATG4C*, *LRR34* and *C11ORF94* as candidates from Table 2 (they were previously 'weakly suggestive') as they no longer passed our multiple testing correction after we applied some more stringent filters requested by reviewer 2.

We began by simply counting the number of diagnosed probands in each cohort who had multi-gene diagnoses. In DDD, we previously reported (Wright et al., NEJM, 2023) that 121 (2.7%) of the 4484 probands who had received a diagnosis by means of clinical assertion had two or more different genetic diagnoses, and that when we included computationally predicted diagnoses (which implement the ACMG criteria automatically), this proportion increased to 359 out of 5502 (6.5%) probands. In GeneDx, which relies on clinicians confirming which candidate variants are pathogenic, 237 out of 9949 diagnosed probands received a composite diagnosis (2.4%), so very similar to the 2.7% in DDD.

We suspected that these numbers might under-estimate the true number of multi-gene diagnoses within the cohorts, since phenotypic heterogeneity conferred by variants in the same gene may make it difficult to assess putatively damaging variants clinically. Thus, to investigate this further, we carried out exome-wide burden analyses of autosomal recessive and *de novo* variants to see whether there was any excess of these in diagnosed patients with single diagnoses (i.e., excluding known or predicted composite diagnoses) once the known diagnostic variants were removed (**Supplementary Figure 15**). We found no significant burden of damaging biallelic variants in probands with a *de novo* diagnosis or in probands with a recessive diagnosis once the diagnostic variants had been removed (attributable fractions 0.1% [-0.5-0.8%] and 0% [-7.6-8.7] respectively). However, we found a significant burden of damaging *de novo* mutations in patients with a *de novo* diagnosis, even after removing the diagnostic variants (attributable fraction 12.5% [9.6-15.5%]; ~502 individuals), as well as in patients with a diagnosis involving inherited dominant, recessive or X-linked variants (attributable fraction 12.5% [8.3-16.8%]; ~241 individuals). Almost all of the residual *de novo* burden in diagnosed individuals was outside of known monoallelic or X-linked dominant DDG2P genes (11.6% [9.3-14.0%], **Supplementary Figure 15**). In DDD, we split the diagnosed patients into those classified by clinicians as having a full *versus* partial diagnosis, and noted that the residual burden of *de novo* mutations (after removing the diagnostic ones) was significantly different from 0 in the ‘fully diagnosed’ set (attributable fraction 5.7% [0.8-10.8%]) but was higher in the partially diagnosed set (attributable fraction 18.0% [3.4-34.5%]).

We have added these results into the manuscript at lines 259-285. We have also added the following discussion at lines 456-470: “Finally, our burden analyses conducted in patients who already have a single genetic diagnosis imply that in as many as ~12.5% of these (~743), an as-yet-unidentified *de novo* mutation in another gene also contributes to the phenotype; almost all of this burden is outside known monoallelic and X-linked dominant DDG2P genes (**Supplementary Figure 15**). If these contributing *de novo* mutations could be identified, it would more than double the number of patients in these cohorts that currently have a composite genetic diagnosis (N=596). We also find that these as-yet-undetected composite diagnoses are more likely amongst patients whose current single diagnosis is deemed ‘partial’ than in those in whom it is deemed ‘full’ (attributable fraction 18% versus 5.7%), and that recessive variants are unlikely to contribute to further composite diagnoses. The 12.5% estimate is higher than previous

estimates of the rate of composite diagnoses³² and this may be for several reasons. Firstly, the excess burden in patients who currently have a single diagnosis may not only reflect dual diagnoses, but may also partly reflect digenic/oligogenic causes, where the second variant may play a role but be insufficient on its own to cause disease. Secondly, as noted below, it may reflect ascertainment bias into the DDD and GeneDx cohorts.”

3. Gene discovery. Table 2 lists the nine genes not on any known AR DD list, but that pass Bonferroni correction.

a. Can the “level of evidence” information be defined by more objective criteria than “very strong”, “strongly suggestive” or “weakly suggestive”?

We have now removed the “level of evidence” column and instead discuss our level of confidence in these genes in the main text (lines 474-478): “Overall, we believe there is strong evidence that *CRELD1*, *KBDBT2*, *ZDHHC16* and *HECTD4* are *bona fide* ARDD genes, whereas the current evidence for *ATAD2B* is more equivocal (**Supplementary Note**).” Note that all but one (*ATAD2B*) of the previously ‘weakly suggestive’ genes were dropped anyway once we implemented stricter allele frequency filters, as suggested by reviewer 2.

b. Further information discussing *CRELD1* is warranted, particularly explanation of how dominant variants give rise to a markedly different phenotype than AR variants.

We have made various changes to our original paragraph about *CRELD1*, including adding some more information about what is known about the gene from mouse studies (lines 319-322), and about the phenotypes described in the Jeffries et al. paper which came out a few months ago reporting this same recessive disorder (lines 327-329). We have also added the following new paragraph (lines 332-347):

“We subsequently identified a further four GeneDx patients (not in the dataset used for our main analysis) with biallelic damaging variants in *CRELD1* that passed our filters; of these, two had phenotypes consistent with the other patients (**Supplementary Table 7**) whereas two did not (see **Supplementary Note**). Of our ten patients with biallelic *CRELD1* variants that we think are likely to be causal, all have genotypes involving missense variants that disrupt cysteine residues in or around the EGF-like and calcium-binding EGF-like domains, so they may destroy disulfide bonding, as noted in Jeffries *et al.*²³; specifically, seven have genotypes involving the p.Cys192Tyr variant previously reported as recurrent²³ (of which two were

previously reported in that paper), one has p.Cys262Arg *in trans* with a pLoF (patient also reported in ²³), and two siblings are homozygous for p.Cys218Tyr. The four additional missense variants reported in Jeffries *et al.* were all in the transmembrane domains at the C-terminal end of the protein. In contrast, the missense variants that have been reported to predispose to AVSD do not show any particular spatial clustering, and none involve cysteine residues. Further functional work would be required to definitively establish the molecular consequences of missense variants contributing to the recessive *CRELD1* neurodevelopmental disorder versus those predisposing to AVSD in the heterozygous state.”

We feel that functional experiments to explore differences between the dominant and recessive variants would be beyond the scope of this paper, since we want to keep the main focus on genetic architecture rather than specific genes.

c. Nearly half of genes in Table 2 are “weakly suggestive” (n=4), and this is disappointing given that these are highlighted among the most likely new causal ARDD genes to come out of these analyses. Further evidence in support of these genes as new ARDD loci is needed (all of them with 2 affected individuals or less), whether it be in the form of additional cases (identified outside of the DDD or GeneDx datasets through gene matching); or through the use of vertebrate animal models to show relevance to DD phenotypes.

Three of the four ‘weakly suggestive’ genes were removed after we applied more stringent filters suggested by reviewer 2. The only one remaining is *ATAD2B*, of which more below.

Through querying an additional 141,417 patients with a neurodevelopmental disorder from GeneDx (who were sequenced after the data freeze on which our main analysis was based) plus >400,000 patients with rare disorders from CentoGene, we have been able to identify several additional cases with likely damaging biallelic variants in the remaining genes in Table 2. These include:

- Two additional *CRELD1* cases from GeneDx. This gene already passed our original Bonferroni correction and was reported recently by Jeffries et al.
- One additional *KBTBD2* case from CentoGene. This gene already passed Bonferroni correction with the two original cases, so the statistical evidence in its favour is already very strong.
- Three new *ZDHHC16* cases, of which two from GeneDx and one from CentoGene. We note that this gene did pass Bonferroni correction when analysing only undiagnosed probands, so has strong statistical evidence even based on only the three initial cases reported.

- Two new *HECTD4* cases from GeneDx. (We note this gene was recently published [Faqeih et al., Genet. Med., 2023].)
- One additional case with rare biallelic missense variants in *ATAD2B* from GeneDx.

In most cases, the phenotypes of these patients were compellingly similar to the patients identified earlier. There is also existing supporting evidence from a mouse model for *KBTBD2* (lines 354-357) and for a zebrafish model for *ZDHHC16* (lines 1341-1343), cited in the text. Taken together, we believe that evidence supporting *KBTBD2* and *ZDHHC16* as novel ARDD genes is strong, and that our data further strengthen the evidence for recently-reported genes *CRELD1* and *HECTD4*.

In the case of *ATAD2B*, there was no particularly striking phenotypic similarity between the patients. Although there is quite a lot of circumstantial evidence supporting this gene, we believe more evidence is needed to confirm it as a real ARDD gene (as noted now at Supplementary Note lines 1387-1402).

Information about these new cases and the genes has been added to Table 2, Supplementary Table 7, the main text (lines 332-335 for *CRELD1*; 364-369 for *KBTBD2*), and the Supplementary Note (lines 1286-1309 for *KBTBD2*; 1318-1336 for *ZDHHC16*; 1351-1355 for *HECTD4*; 1366-1370 for *ATAD2B*).

d. Phenotype similarity is difficult to appreciate for individuals listed in Supplementary Table 7. It would be extremely helpful to have individual HPO terms listed by column and indicators of present/absent/no data for each individual so that overlap can be better visualized.

We have now done as the reviewer has asked. Unfortunately, for most cases, we do not have the information to distinguish between “absent” and “no data”, so the table just indicates whether the HPO term (or a descendent) was present. In cases where we were able to specifically check with the clinician that a phenotypic feature was absent, we have indicated this in the “Additional info/phenotypes” column. Additionally, we do not have permission to list the precise HPO terms for most patients, so instead have just listed chapter-level HPO terms.

Reviewer #2:

Remarks to the Author:

With this study, the authors expand on a previous one, published in 2018, where they estimated the genome-wide contribution of recessive variants in 6040 families from the Deciphering Developmental

Disorders study. The same objective led them to combine this data with new one from patients of the genetic diagnostics company GeneDx who had some “abnormality of the nervous system,” thus creating a new cohort of 29,745 patients. They created genetically inferred ancestry (GIA) groups and sub-groups with little significant outcome. They identified 2 novel autosomal recessive developmental disorders (ARDD) genes, *KBTBD2* and *CRELD1*, and found suggestive evidence for *ZDHHC16*, *ATG4C*, *HECTD4*, *ATAD2B*, *ATXN1*, *LRRC34*, and *C11ORF94*. They also highlighted the challenges of such approach.

I have major and minor comments.

It would be useful if the authors discussed the function of the proteins, and ideally the expression pattern, encoded by the genes they suggest are new recessive disease genes. For example, *C11ORF94* seems important for spermatids and seems expressed specifically in the testes, this raises concerns for its implication in DD. More on this gene in the following comment.

We have added some text about the functions and expression patterns of the genes in Table 2 into the main Results and the Supplementary Note in the following places:

- *CRELD1*: lines 319-324
- *KBTBD2*: lines 350-356; 362-364
- *ZDHHC16*: lines 1341-1343
- *HECTD4*: lines 1357-1361
- *ATAD2B*: lines 1387-1391

Please note that *C11ORF94* and two of the other genes have been dropped due to the new filters described below.

Major comments:

I am concerned by the stringency of the filtering for the variants. For example, *ATG4C* is tolerant to LoF variants. Variant chr1-63299729-TTG-T found in 5 of 7 patients, some at homozygosity, is carried by 1/70 European, and is found at homozygosity in multiple individuals in the gnomAD v2.1.1 non-neuro cohort. Variants in *CRELD1*, *LRRC34* and *C11ORF94* especially also have non-negligible MAFs in gnomAD and such variants are present in 50% or more of the patients for these 3 genes. For example, *C11ORF94* is tolerant to LoF variants also, and one of the two variants (found at homozygosity in 1 of 2 patients), chr11-

45928455-AG-AGG, when searched with a different nomenclature (chr11-45928455-A-AG), is found in about 1/300 Latinos.

The reviewer raises a valid point. In hindsight, our initial filtering (MAF<1%) was probably too lenient. We have now revised our analyses to remove variants found in the homozygous state in any gnomAD individuals, or with minor allele frequency >0.5%. We chose this MAF cutoff since simulations based on a plausible demographic model for European populations suggests that recessive, reproductively lethal mutations (which most variants causing these severe disorders effectively are) can rise to frequencies of ~0.5% by chance (Amorim et al., PLoS Genetics, 2017). Implementing this new MAF cutoff dropped ~3% of recessive variants annotated as pathogenic by clinicians in our combined DDD/GeneDx dataset. These new filters made very little difference to our estimates of exome-wide burden or any of the estimates presented in Figure 1 or 2. The few results which changed quantitatively were:

- that the total AR attributable fraction was no longer significantly different between GeneDx and DDD (4.1% versus 3.8%, p=0.23)
- that the fraction of exome-wide AR burden due to consensus and discordant genes became significantly different between patients with European and non-European ancestries, although this difference was relatively small in magnitude (86.9% versus 79.8%, p=0.003)

However, these new filters did reduce the significance for three of the four genes we had previously called 'weakly suggestive' (*ATG4C*, *LRCCC34* and *C11ORF94*), so they no longer crossed the FDR<5% thresholds and have been dropped from Table 2. Since there were various reasons these genes were previously not very convincing, it is satisfying to see that they drop out with this new filtering. We note that the two *CRELD1* variants occurring in multiple cases as part of compound heterozygous genotypes (chr3:9982648:G>A and chr3:9985109:CA>C) both pass these new filters and have a maximum frequency in any gnomAD population of <0.0006, which is reasonable for a recessive variant.

As an aside, we do note that truly recessive genes often are tolerant to *heterozygous* LoFs, and hence have a low pLI score. The original ExAC paper showed that the pLI score is not very informative about recessive genes (Supplementary Figure 4 of Lek et al., Nature, 2016). However, we contend that the converse is not true - if a gene *does* have a high pLI, but loss-of-function variants in it *do not* cause a severe dominant disorder (as is the case for several of the genes in Table 2, *KBTBD2*, *HECTD4* and *ATAD2B*), it seems more likely to cause a recessive disorder than a gene with low pLI.

Minor comments:

“We estimated that ~1% of undiagnosed patients in both cohorts (...): As the reader assumes that the authors are referring to the probands with genetically inferred European vs. non-European ancestries, this population segregation is not previously explained nor justified, especially considering that poorly significant results were obtained from GIA comparisons.

Apologies, in fact “in both cohorts” here referred to GeneDx and DDD, not to two groups with different ancestries. We have just removed “in both cohorts” from this line in the abstract.

“We defined six continental-level GIA groups (...) and, within these, forty-seven fine-scale GIA sub-groups”: It is important to give some detail regarding the GIA sub-groups (e.g. based on a certain surface area, or population density, or ethnic distribution, or allele frequencies or autozygosity levels) elsewhere than in the supplementary data.

We have modified this section in the Results to make this clearer: “The classifications were based on genetic similarity to individuals in the 1000 Genomes and Human Genome Diversity Panel (HGDP) reference datasets, inferred from principal component analysis (Supplementary Figures 3 and 4). **Using statistical clustering of individuals based on their genotypes**, we defined six continental-level GIA groups...” (lines 107-111). Table 1 indicates the closest corresponding reference population to each GIA sub-group, and the full details are described in the main Methods section under “Ancestry assignment”. We feel that is sufficient detail for the main text but if the reviewer or editor feels strongly, we can add more detail into this paragraph of the Results.

“This study (...) suggests that improving strategies for interpreting missense variants in known ARDD genes may allow us to diagnose more patients than discovering the remaining genes”: This conclusion seems quite evident. I would talk in addition about the limitations of a cohort of patients with complicated diagnosis, or relatively unselected cohorts, or other elements specific to this study’s cohort.

We suspect the reviewer means that it’s evident that better interpreting missense variants in known ARDD genes would be generally a good thing. We certainly agree with this, but we don’t believe that it was evident *a priori* that better interpreting missense variants in known ARDD genes would diagnose more DD patients than discovering the remaining ARDD genes. This conclusion is only possible to make through the kind of large-scale statistical burden analysis we have done, thus we feel it is an important point to

emphasize in the abstract. (The numbers backing up this statement are given in the Discussion, lines 448-453.)

Since there is no space in the abstract, we have added in a mention of the other limitations raised by the reviewer at lines 484-489: “Firstly, the families studied are not a random sample of the DD patient population, and may be depleted of easy-to-solve families with recessive conditions. Thus, we may have underestimated the contribution of AR variants to DDs as a whole, **overestimated the true rate of composite diagnoses, or overestimated the overall fraction of new diagnoses that could be made by better interpreting missense variants in known ARDD genes.** “

“clinically recognizable disorder (e.g.11,12)”: I would either say “clinically recognizable disorder 11,12” or “clinically recognizable disorder (e.g. Miller11 and Wiedemann-Steiner12 syndromes)”

We have changed it to the latter (line 63).

“Forty-eight percent of the exome-wide burden of recessive causes was in known AR DD-associated (ARDD) genes, indicating that larger sample sizes would be required to find the additional genes”: As your study shows, larger cohorts are not necessarily more useful than better-curated cohorts.

We agree that larger cohorts of the same type as the one we have are not an efficient strategy. In our opinion, enriching cohorts for consanguineous families with multiple affected family members (and recruiting those family members) is likely to boost power for gene discovery, as is focusing on founder populations. We have changed this sentence (lines 75-76) to “indicating that larger sample sizes **and/or a different study design** would be required to find the additional genes”. In the Discussion we mention that “discovery of the remaining ARDD genes will require larger samples and/or more focused sampling of genetically isolated communities enriched for causal founder variants and/or consanguineous families with multiple affected individuals” (line 505-507).

“while the attributable fraction due to de novos was not”: “de novo variants” or “DNMs” instead

Thanks for pointing this out. It is now corrected (line 173).

Decision Letter, first revision:

12th June 2024

Dear Hilary,

Your revised manuscript "Federated analysis of the contribution of autosomal recessive coding variants to 29,745 developmental disorder patients from diverse populations" (NG-A63077R) has been seen by the original referees. As you will see from their comments below, they find that the paper has improved in revision, and therefore we will be happy in principle to publish it in Nature Genetics as an Article pending final revisions to comply with our editorial and formatting guidelines.

We are now performing detailed checks on your paper, and we will send you a checklist detailing our editorial and formatting requirements soon. Please do not upload the final materials or make any revisions until you receive this additional information from us.

Thank you again for your interest in Nature Genetics. Please do not hesitate to contact me if you have any questions.

Sincerely,
Kyle

Kyle Vogan, PhD
Senior Editor
Nature Genetics
<https://orcid.org/0000-0001-9565-9665>

Reviewer #1 (Remarks to the Author):

I remain highly enthusiastic about this study, which characterizes the contribution of autosomal recessive (AR) coding variants to disease in nearly 30k parent proband-trios from the DDD and GeneDx cohorts. The revision contains a new analysis which reports multi-gene diagnoses per my previous suggestion, and I was delighted to see this incorporated into the paper in the main text and Supplementary Figure 15. Another important modification to this study includes more robust analyses to remove variants found in the homozygous state in any gnomAD individuals or with MAF >0.05%.

The work remains highly original and significant with conclusions drawn from robust approaches. However, in this version, the authors did not include variants on the X-chromosome (per my previous suggestion). In lieu of performing this analysis, which I understand could not take place for largely technical reasons, the authors have clarified throughout the manuscript that their focus was on autosomal variation, which I think is fair. I agree with their point that such an analysis is unlikely to yield substantial new insights about the X-linked contribution to DD beyond those made in their 2021 Nature Communications paper and am willing to accept the paper in its current form.

Reviewer #2 (Remarks to the Author):

All my concerns were appropriately addressed.

Final Decision Letter:

14th Aug 2024

Dear Dr Martin,

I am delighted to say that your manuscript "Federated analysis of autosomal recessive coding variants in 29,745 developmental disorder patients from diverse populations" has been accepted for publication in an upcoming issue of Nature Genetics.

Your paper will be published online after we receive your corrections and will appear in print in the next available issue. You can find out your date of online publication by contacting the Nature Press Office (press@nature.com) after sending your e-proof corrections.

Before your paper is published online, we shall be distributing a press release to news organizations worldwide, which may very well include details of your work. We are happy for your institution or

funding agency to prepare its own press release, but it must mention the embargo date and Nature Genetics. Our Press Office may contact you closer to the time of publication, but if you or your Press Office have any enquiries in the meantime, please contact press@nature.com.

Please note that *Nature Genetics* is a Transformative Journal (TJ). Authors may publish their research with us through the traditional subscription access route or make their paper immediately open access through payment of an article-processing charge (APC). Authors will not be required to make a final decision about access to their article until it has been accepted. Find out more about Transformative Journals

Authors may need to take specific actions to achieve compliance with funder and institutional open access mandates. If your research is supported by a funder that requires immediate open access (e.g. according to Plan S principles) then you should select the gold OA route, and we will direct you to the compliant route where possible. For authors selecting the subscription publication route, the journal's standard licensing terms will need to be accepted, including [a href="https://www.nature.com/nature-portfolio/editorial-policies/self-archiving-and-license-to-publish](https://www.nature.com/nature-portfolio/editorial-policies/self-archiving-and-license-to-publish). Those licensing terms will supersede any other terms that the author or any third party may assert apply to any version of the manuscript.

If you have not already done so, we strongly recommend that you upload the step-by-step protocols used in this manuscript to protocols.io. protocols.io is an open online resource that allows researchers

to share their detailed experimental know-how. All uploaded protocols are made freely available and are assigned DOIs for ease of citation. Protocols can be linked to any publications in which they are used and will be linked to from your article. You can also establish a dedicated workspace to collect all your lab Protocols. By uploading your Protocols to protocols.io, you are enabling researchers to more readily reproduce or adapt the methodology you use, as well as increasing the visibility of your protocols and papers. Upload your Protocols at <https://protocols.io>. Further information can be found at <https://www.protocols.io/help/publish-articles>.

Sincerely,

Safia Danovi, PhD
Senior Editor, Nature Genetics
ORCID: 0009-0007-7822-5479